# Reconfigurable perovskite X-ray detector for intelligent imaging

Jincong Pang[1,3], Haodi Wu[1,3], Hao Li[1], Tong Jin[1], Jiang Tang [1,2] & Guangda Niu [1,2] ✉

X-ray detection is widely used in various applications. However, to meet the demand for high image quality and high accuracy diagnosis, the raw data increases and imposes challenges for conventional X-ray detection hardware regarding data transmission and power consumption. To tackle these issues, we present a scheme of in-X-ray-detector computing based on $CsPbBr_3$ single-crystal detector with convenient polarity reconfigurability, good linear dynamic range, and robust stability. The detector features a stable trap-free device structure and achieves a high linear dynamic range of 106 dB. As a result, the detector could achieve edge extraction imaging with a data compression ratio of ~50%, and could also be programmed and trained to perform pattern recognition tasks with a high accuracy of 100%. Our research shows that in-X-ray-detector computing can be used in flexible and complex scenarios, making it a promising platform for intelligent X-ray imaging.

X-ray detection is crucial in multiple areas, like detecting radioactive materials, inspecting industrial flaws, security screening, and medical imaging[1–5]. Over the years, significant advancements have been made to X-ray detectors, leading to improved sensitivity, superior image contrast, fast response speed, and high spatial resolution. However, these advancements have also resulted in a substantial increase in data volume, posing challenges in data transmission, processing, and storage[6–10]. For instance, high-end computed tomography (CT) detectors generate data at a rate exceeding 40 gigabits per second (Gbps), an output precipitated by factors like increased rotating speed, slice counts, and detector rows. This outpaces the capabilities of the transmission unit (CT slip ring, with an approximate limit of 10 Gbps), arithmetic unit (reconstruction rate of fewer than 64 images per second), and storage unit. Similarly, dynamic digital radiography yields data volumes as massive as tens to hundreds of Gbps, exceeding the transmission abilities of interfaces like CameraLink HS (10 Gbps) and GigE (6.8 Gbps). There have been attempts to bolster the transmission, processing, and storage rates through methods like photoelectric coupling, multiplexing, interface enhancements, and GPU upgrades. However, the progress achieved so far has been limited (Supplementary Tables 1 and 2). Additionally, traditional X-ray

detection systems, which depend on separate sensors, analog-to-digital converters, and processing modules, are plagued by issues such as excessive bulk, high complexity, and considerable power consumption.

The emerging technology of in-sensor computing, also referred to as neural network vision sensors, has demonstrated its potential in achieving low power consumption, minimal latency, and overcoming transmission bandwidth limitations in visible light detection[11–13]. Inspired by these advancements, the adoption of in-sensor computing architecture holds promise in mitigating the above limitations of conventional X-ray detection systems. Nevertheless, the application of similar in-sensor computing capabilities in X-ray detectors has not yet been reported.

The in-sensor computing architecture requires detectors with certain characteristics of polarity reconfigurability and linear responsivity[11–14], which are challenging to achieve using conventional X-ray detectors. For example, scintillator-based indirect detectors could hardly be reconfigured. Direct detectors based on a-Se have low sensitivity and poor response at low doses[15], while Cd(Zn)Te suffers from poor hole collection capability and limited dynamic range at reverse bias[16]. In contrast, halide perovskites have emerged as

[1]Wuhan National Laboratory for Optoelectronics and School of Optical and Electronic Information, Huazhong University of Science and Technology, 430074 Wuhan, China. [2]Optical Valley Laboratory, 430074 Wuhan, China. [3]These authors contributed equally: Jincong Pang, Haodi Wu. ✉e-mail: guangda_niu@hust.edu.cn

excellent semiconductors for direct X-ray detection[17]. They possess several desirable properties including bias-tunable response, high sensitivity, low detection limit, and balanced charge collection, making them well-suited for developing X-ray detectors with in-sensor computing capability.

Here, we report the in-X-ray-detector computing based on an N-I-P type $CsPbBr_3$ perovskite single-crystal (PSC) detector with convenient polarity reconfigurability, good linear dynamic range as well as robust stability. To address the common issue of poor linearity observed in perovskite semiconductors, we extensively analyze the traps at the interface of single crystals, identify them as the root cause, and effectively passivate the surfaces. As a result, we achieve a high linear dynamic range of 106 dB. Furthermore, we harness the fabricated X-ray detector's ability to perform intelligent edge extraction for data compression, achieving an impressive compression ratio of ~50%. Additionally, the constructed X-ray detector has the capability to concurrently sense and process images with various kernels. It is also shown that this detector can be trained for pattern recognition tasks, achieving an impressive accuracy rate of 100%. The approach presented in this research paves the way for future applications of advanced neural network-based X-ray detectors.

## Results

### The principle of imaging based on convolution kernel

As mentioned earlier, the large amount of data generated by X-ray imaging is becoming a bottleneck in data transmission. We use CT as a case to illustrate the rapid increase of the data generated by detectors, which is calculated by the method in Supplementary Note 1 and data in Supplementary Tables 1 and 2. Our design concept involves the construction of a macro-pixel composed of several sub-pixels, and the combination of convolution operations to derive pre-processed images. This diverges from the traditional X-ray detection architecture, which transmits signals to a computing unit for processing. The simultaneous detection of X-ray photons and preliminary data processing can be achieved through a single readout procedure, as depicted in the operating mechanism illustrated in Fig. 1a. Based on Kirchhoff's current law, the macro-pixel's output signals are represented as $\sum_j R_j$, with $R_j$ denoting the sub-pixel's signal value. The computation within the X-ray detector becomes feasible by setting the $R_j$ values to specific positive and negative amounts determined by the bias polarity.

Figure 1a illustrates the data processing through specific convolutional kernels to achieve the effect of edge enhancement, contrast correction, and data compression. In detail, the central sub-pixel in the macro-pixel is applied with a forward bias value, and the surrounding sub-pixels are applied with reverse bias values according to the Laplacian convolution kernel to obtain negative signals under the same X-ray intensity, which are one-eighth of the positive signal. Therefore, it outputs the off state (zero signal) for the position with uniform contrast and outputs the on state (non-zero signal) for the position with changed contrast. From the perspective of mathematics, the aforementioned operation is equal to the derivation of the image, leading to the edge enhancement effect. Depending on imaging requirements, different convolutional kernels can be achieved by setting the weight matrix of the sub-pixels, resulting in varying compression ratios. For example, the Laplacian of the Gaussian (LoG) convolutional kernel depicted in Fig. 1a, can achieve a compression ratio of 49.4% (636,164 effective pixels in a 1430 × 900 grid); whereas

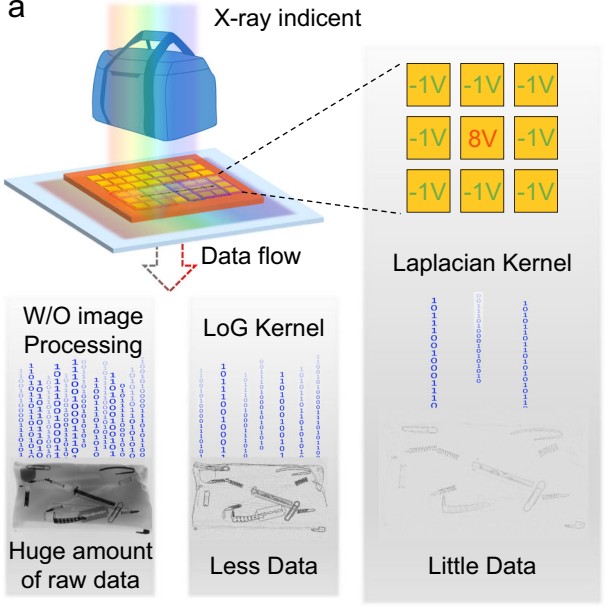
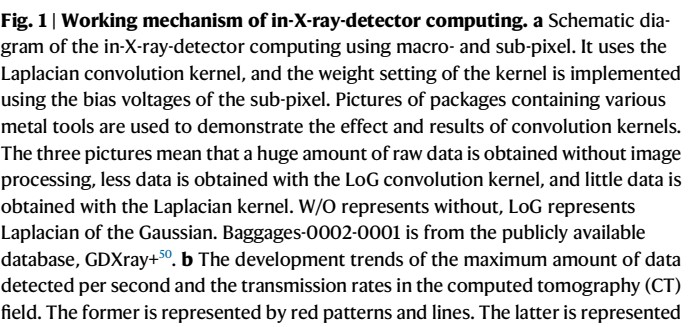
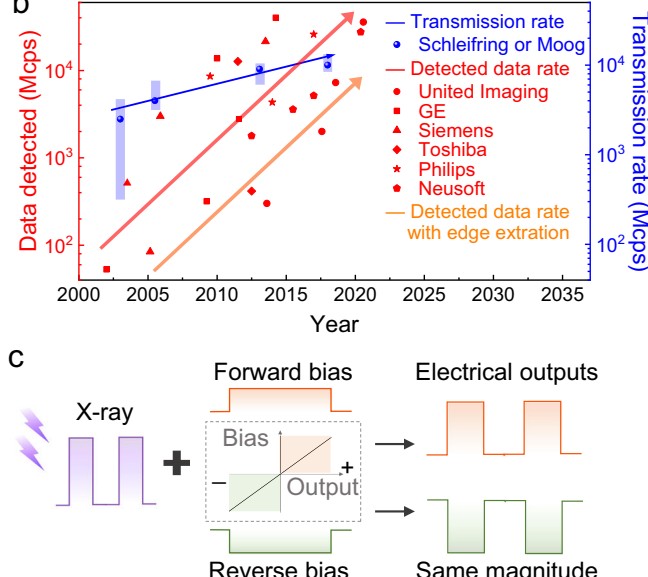

**Fig. 1 | Working mechanism of in-X-ray-detector computing. a** Schematic diagram of the in-X-ray-detector computing using macro- and sub-pixel. It uses the Laplacian convolution kernel, and the weight setting of the kernel is implemented using the bias voltages of the sub-pixel. Pictures of packages containing various metal tools are used to demonstrate the effect and results of convolution kernels. The three pictures mean that a huge amount of raw data is obtained without image processing, less data is obtained with the LoG convolution kernel, and little data is obtained with the Laplacian kernel. W/O represents without, LoG represents Laplacian of the Gaussian. Baggages-0002-0001 is from the publicly available database, GDXray+[50]. **b** The development trends of the maximum amount of data detected per second and the transmission rates in the computed tomography (CT) field. The former is represented by red patterns and lines. The latter is represented by blue patterns and lines. Among them, products from United Image are represented by circles, products from General Electric Company (GE) are represented by squares, products from Siemens are represented by triangles, products from Toshiba are represented by rhombus, products from Philips are represented by stars, and products from Neusoft are represented by pentagons. Schleifring and Moog's products use rectangles to delineate ranges of rates, and dots to represent typical values. The orange line is the compressed detected data rate after implementing the convolutional kernel in this work. Details can be found in Supplementary Note 1, and Tables 1 and 2. **c** The bias-tunable X-ray response with the same magnitude of perovskite detectors. This is the premise of the in-X-ray-detector computing architecture.

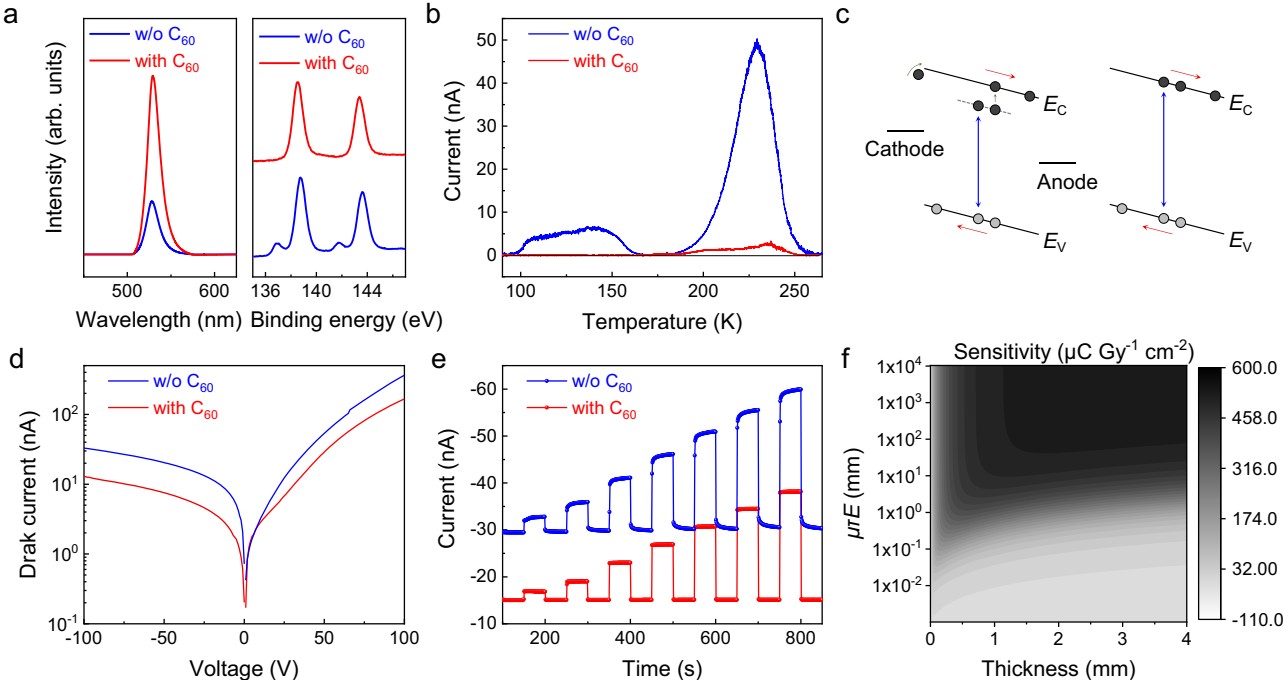

**Fig. 2 | Study of defects in PSC with and without the C$_{60}$ passivation layer.**
**a** X-ray photoelectron spectroscopy (XPS) and photoluminescence (PL) spectra.
**b** Thermally stimulated current (TSC) curves. The results of split-peak fitting can be viewed in Supplementary Fig. 4 and the method can be viewed in Supplementary Note 2. **c** Schematic diagram of photoconductive gain effect with and without defect energy levels. The defect level traps one type of photogenerated carrier.

To maintain the neutrality of the semiconductor, the electrodes inject carriers. This will result in the photoconductive gain. **d, e** I–V and I–t curve. w/o represents without. **f** Simulation values of the intrinsic sensitivity with various thicknesses and $\mu\tau E$ for the perovskite single crystal (PSC). Details can be found in Supplementary Note 3. Source data are provided as a Source Data file.

the Laplacian convolution kernel yields an improved compression ratio of 46.6% (600,261 effective pixels).

Figure 1b details the data volume and transmission rate bottlenecks of CT products. The edge enhancement, or in-X-ray-detector computing technology represented by the orange straight line, provides a potential solution to the data overload issue. The in-X-ray-detector computing architecture requires detectors with the characteristic of polarity reconfigurability. It is shown in Fig. 1c, that the kernel value of each pixel can be tuned by the bias voltage and the perovskite X-ray detector shows a bias-tunable linear response. This will be explained in detail later.

## High linear dynamic range single-crystal detector

To achieve the above target, we have to address the problem of poor linearity of perovskite semiconductors, which has been observed in many recent works[18–23]. Physically speaking, the deep reason is expected from the presence of traps and thus polarization effect under high flux, and sublinear trap-limited photo-response effect under the low flux of X-ray[24–26]. Here we select Bridgeman-grown CsPbBr$_3$ single crystal for study. To guarantee the quality of crystal bulk, we purify the raw materials by zone-melting multiple times to a purity of around 99.9999%, as shown in Supplementary Table 3. This reduces defects caused by impurity elements. The as-synthesized ingot is then cut and polished to obtain the sample.

We fabricate an N-I-P type device Bi-ZnO-CsPbBr$_3$-NiO$_x$-C, where inorganic transport layers are selected to ensure the high bias stability and the details can be found in the Methods section and Supplementary Fig. 1. However, after directly depositing the inorganic transport layers, we often encounter device failures and poor linearity response. To understand the chemical origin, we obtain the X-ray photoelectron spectroscopy (XPS) of Pb 4$f$, and find that there were additional peaks at 136.9 eV and 141.7 eV besides the binding energy peak for Pb$^{2+}$ (138.7 and 143.6 eV), as shown in Fig. 2a. The two additional peaks belong to

undercoordinated Pb defects (Pb$^0$ state) on the surface. The photoluminescence intensity is rather low. The presence of undercoordinated Pb defects is probably due to surface damage from the magnetron sputtering or polishing process and thus halide vacancies. Consequently, we introduce an additional C$_{60}$ layer, which has been proven to be an effective passivator, to suppress the surface states[27–30]. The additional peaks of undercoordinate Pb in XPS disappeared and the photoluminescence intensity is significantly boosted in Fig. 2a. Characterization of the C$_{60}$ layer can be seen in Supplementary Fig. 2 and control experiments under different sputtering powers can be seen in Supplementary Fig. 3.

We also quantitatively characterize the defect depth through the thermally stimulated current spectrum (TSC). Before passivation, high-intensity currents are detected at 110-135 K and 225 K, corresponding to defect depths of 0.27 eV and 0.48 eV, respectively. The defect density is estimated as $1.24 \times 10^{14}$ cm$^{-3}$ and $5.00 \times 10^{14}$ cm$^{-3}$, in Fig. 2b, aligning closely with the results for perovskite single crystals in other studies about CsPbBr$_3$, such as $3.84 \times 10^{14}$ cm$^{-3}$, $4.11$-$52.9 \times 10^{14}$ cm$^{-3}$, or $1.82$-$68.1 \times 10^{14}$ cm$^{-3}$ [31–33]. According to the previous study, the defects are probably from the Pb-Pb dimers[34], which formed on the surface as a consequence of halide vacancies[35,36]. After passivation, the device shows negligible defects in the whole range. Detailed simultaneous multiple peak analysis results and defect concentration calculation can be found in Supplementary Note 2 and Supplementary Fig. 4. The presence of defects can trap photogenerated carriers and result in the photoconductive gain effect[24–26], as shown in Fig. 2c. The gain value highly depends on the photogenerated carrier density, which is proportional to the intensity of X-ray photons. The above behavior results in the sublinear response at low intensities, which is common in X-ray detectors and photodetectors. The elimination of surface defects and perfect bulk state theoretically guarantee the linear response.

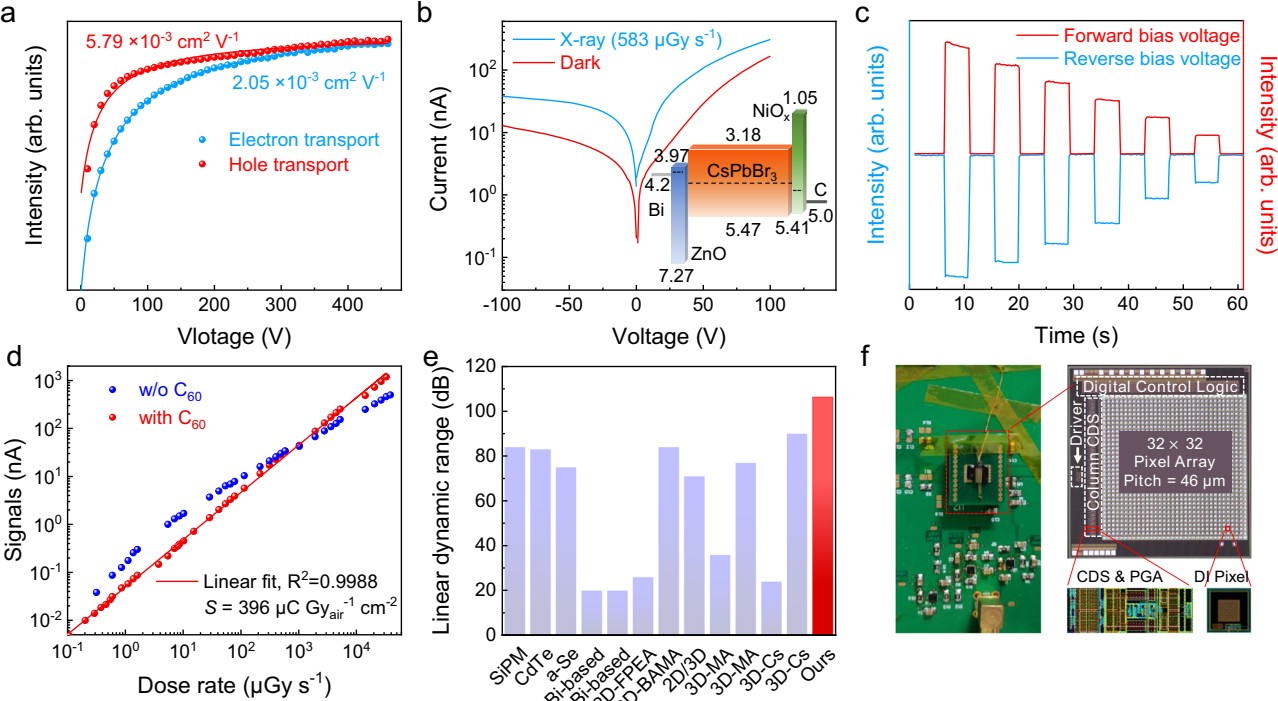

**Fig. 3 | The perovskite single-crystal device performance. a** $\mu\tau$ fitted by the Hecht formula. **b**, **c** The $I-V$ and $I-t$ curves under forward and reverse bias voltages with variable X-ray intensity. The illustration shows the N-I-P perovskite single-crystal (PSC) device structure and the energy bands are obtained according to Supplementary Fig. 1. **d** Linear dynamic range (LDR) results. **e** Comparison of LDR of semiconductor X-ray detectors. **f** The area array detector integrated onto the complementary metal-oxide-semiconductor (CMOS), the optical photo, and the structural diagram of the CMOS die. Detailed schematic diagrams and process photos can be viewed in Supplementary Figs. 8 and 9. CDS represents correlated double sampling, PGA represents programmable gain amplifier, and DI represents direct injection. Source data are provided as a Source Data file.

Then we characterize the electrical properties. The detector with C$_{60}$ passivation exhibits a better rectification effect. The dark current reaches −14 nA at −100 V and 166 nA at 100 V in Fig. 2d, which is among the lowest for CsPbBr$_3$ detectors[31,37–39]. Upon X-ray illumination, the detector displays a good linear response to the dose rates between 32 to 494 μGy s$^{-1}$, a stable baseline, and a fast switching speed under −100 V, as shown in Fig. 2e. The sensitivity reaches 396 μC Gy$_{air}^{-1}$ cm$^{-2}$. It should be noted that this value is lower than previously reported values[37–39], while the previously high sensitivity is typically due to the photoconductive gain effect and thus negatively results in a nonlinear response.

In order to confirm that the sensitivity value is exclusively due to the intrinsic response of the detector, rather than photoconductive gain, we calculate the intrinsic sensitivity of CsPbBr$_3$ by taking into consideration charge collection efficiency and electron-hole pair creation energy[40]. Under our experimental X-ray source (Target: gold; Driving voltage: 70 kVp), the theoretical intrinsic sensitivity of CsPbBr$_3$ is 474 μC Gy$_{air}^{-1}$ cm$^{-2}$ (Fig. 2f), given a mobility-lifetime product ($\mu\tau$) value of 2.05 × 10$^{-3}$ cm$^2$ V$^{-1}$, a bias voltage of 100 V, and a thickness of 2 mm. Details about the simulation method and parameters can be found in Supplementary Note 3. The experimentally measured sensitivity is very close to the intrinsic sensitivity in Fig. 2f, validating that the defects in our device are well-passivated and the device is gain-independent.

Then we systematically evaluate the key performance metrics of the passivated detectors for in-X-ray-detector computing. The mobility-lifetime product, $\mu\tau$ is obtained by fitting the photoconductivity curves using the modified Hecht equation:

$$I = \frac{I_0 \mu\tau V}{L^2} \frac{1 - \exp(-L^2/\mu\tau V)}{1 + Ls/\mu V} \qquad (1)$$

where $I$ is the X-ray-response current, $I_0$ is the saturated photocurrent, $L$ is the thickness, $V$ is the bias voltage, $\mu$ is the carrier mobility, $\tau$ is the carrier lifetime and $s$ is the surface recombination rate.

Here we derive the specific $\mu\tau$ value for electron and hole according to the illumination direction. The $(\mu\tau)_e$ is 2.05 × 10$^{-3}$ cm$^2$ V$^{-1}$, which is close to the $(\mu\tau)_h$ (0.58 × 10$^{-3}$ cm$^2$ V$^{-1}$), as shown in Fig. 3a. In comparison, conventional X-ray detection semiconductors exhibit seriously imbalanced electron/hole $\mu\tau$, as seen in examples like CdZnTe ($(\mu\tau)_e = 7 \times 10^{-3}$ cm$^2$ V$^{-1}$, $(\mu\tau)_h = 9 \times 10^{-5}$ cm$^2$ V$^{-1}$)[16], CdTe ($(\mu\tau)_e = 1.9 \times 10^{-3}$ cm$^2$ V$^{-1}$, $(\mu\tau)_h = 7.5 \times 10^{-5}$ cm$^2$ V$^{-1}$)[41], a-Se ($(\mu\tau)_e = 4 \times 10^{-7}$ cm$^2$ V$^{-1}$, $(\mu\tau)_h = 2 \times 10^{-5}$ cm$^2$ V$^{-1}$)[15], TlBr ($(\mu\tau)_e = 1 \times 10^{-2}$ cm$^2$ V$^{-1}$, $(\mu\tau)_h = 4 \times 10^{-4}$ cm$^2$ V$^{-1}$)[42], as shown in Supplementary Fig. 5a. The $\mu\tau$ product represents the distance that the charges can transit under specific electric field strength, and the balanced $\mu\tau$ values guarantee the response linearity under forward and reverse biases when programmed in the convolutional kernels, based on the analysis in Supplementary Note 4. As shown in Fig. 3b, we record the $I-V$ curves of the detector under both X-ray illumination and the dark state. The response of the detector depends on the polarity of the bias, which is necessary to achieve the as-described convolution kernel. As the X-ray dose rates vary, the signals under forward and reverse bias voltages (95 and −73 V) remain almost the same magnitude and the on-off ratio, as shown in Fig. 3c. This is the prerequisite for in-X-ray-detector computing.

Subsequently, the linearity of the PSC device's response is evaluated. It's important to highlight that this high linearity not only enables the adjustment of convolutional kernels but also aids in the facilitation of weight updates during the convolutional neural network's training phase. Furthermore, it is of more importance for in-X-ray-detector computing devices than for conventional detectors, as mentioned in Supplementary Note 5. In Fig. 3d, the well-passivated

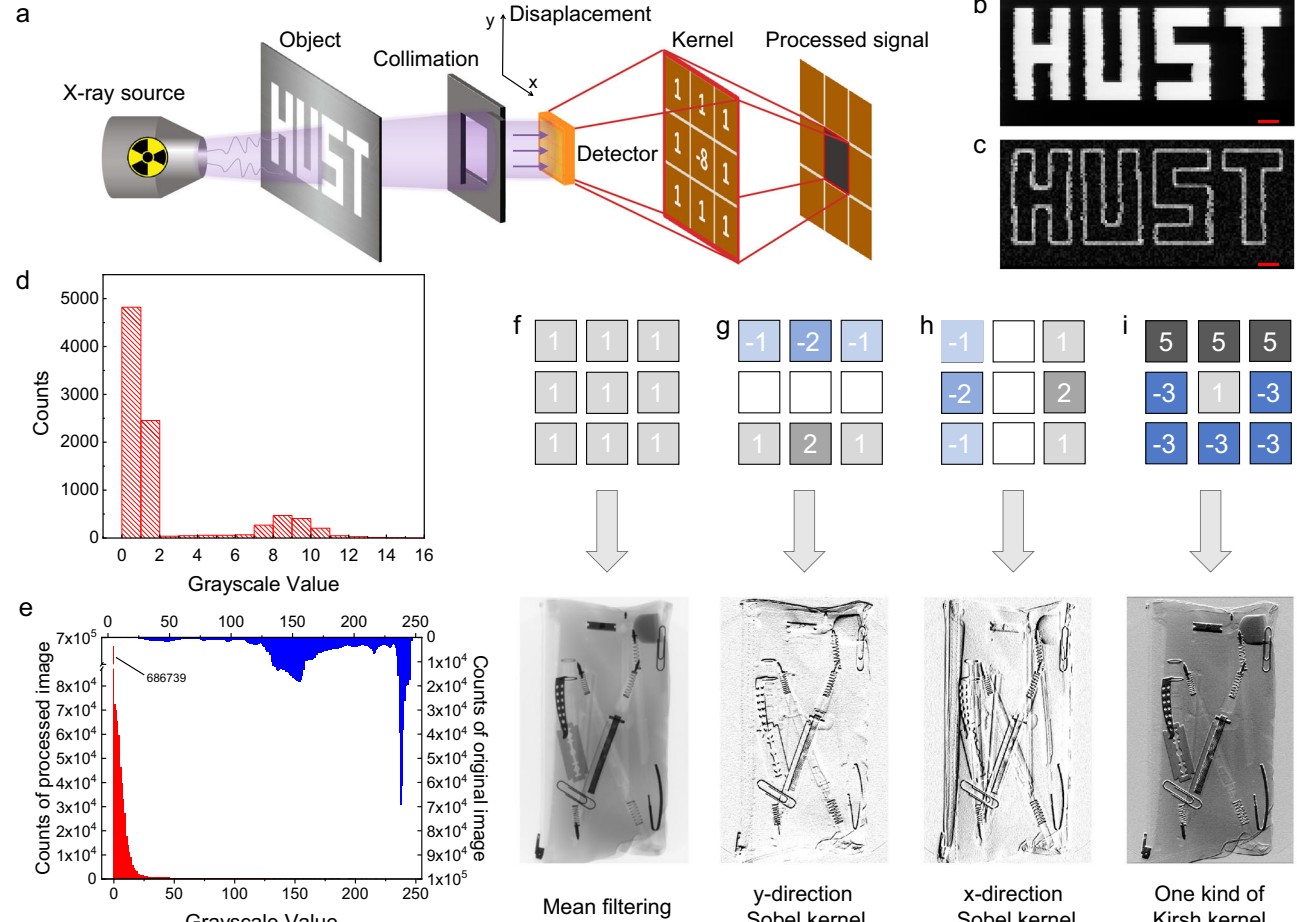

**Fig. 4 | Imaging for edge extraction. a** The schematic diagram of imaging. **b** The X-ray imaging of the experiment, following conventional methods. **c** The imaging following the in-X-ray-detector computing method (the weight values of the pixels are allocated according to the Laplacian convolution kernel). **d** The gray distribution histogram of (**c**). **e** The histogram of the gray distribution of Baggages-0002-0001 with (red) or without (blue) Laplacian simulation processing. In the blue grayscale histogram, which represents the result processed by the convolution kernel, 686,739 pixels have a grayscale value less than 1. **f–i** Other convolution kernels with their pixel weights, and the effects of the simulation image stylization. All scale bars in red color are 1 cm. Source data are provided as a Source Data file.

detector exhibits no trap-limiting sublinear response under low X-ray intensity. Moreover, there is no significant self-limiting response deviation caused by limited electron mobility at high X-ray intensity. The good linearity ranges from 0.165 to 32,920 µGy s$^{-1}$, and the linear dynamic range (LDR) reaches 106 dB. In sharp contrast, the control device without passivation exhibits a trap-limiting gain effect under low dose rates, with a linear dynamic range of merely 68 dB. Here, we compare the documented linear dynamic range for various X-ray detectors. The detector in this study surpassed the previously reported results and state-of-the-art commercial products in Fig. 3e, including SiPM 84 dB[18], CdTe 83 dB (from ANSeeN, CdTe sensor), a-Se 75 dB (from KA Imaging, BrillianSe), and perovskites 20 dB (Bi-based)[19,20], 26-84 dB (2D perovskite)[21,43,44], 24-90 dB (3D perovskite)[22,23,45–47]. In addition, the device exhibits very good robustness, and the performance does not change much under long-term placement and irradiation as shown in Supplementary Fig. 6. The device exhibits relatively fast response and decay times under the irradiation of X-ray and visible light in Supplementary Fig. 7. For the comparison of the response speed between the device and other PSC detectors, readers can view Supplementary Tables 4 and 5. Readers can view Supplementary Table 6 to find the summary performance comparison.

We further integrate the PSC device onto the complementary metal-oxide-semiconductor (CMOS) die using the flip-chip bonding method (see details in Supplementary Figs. 8–10, and Methods). The

optical images of them are shown in Fig. 3f. The area array on it has a total of 32 × 32 pixels, and the outermost circle is used to meet the needs of the process and does not participate in imaging. Previous studies all focus on the integration of perovskite with TFT arrays[48,49]. As far as we are concerned, there is no report on the heterogeneous integration of PSC on CMOS. We note that the low-temperature cured conductive film and the macroscopically flat sample surface are very important for the flip-chip process of PSC and CMOS. These help us achieve high-contact-yield integration.

## The effect of edge extraction and data compression

With the bias-tunable linear X-ray detectors, we can implement convolutional kernels to achieve edge sharpness and data compression effect. For imaging real objects, we utilize a macro-pixel detector with an aforementioned device structure. The test system is depicted in Fig. 4a and Supplementary Fig. 11. A collimator, placed between the target object for imaging and the detector, is a crucial component of the system. After passing through the steel reticle object with a HUST pattern and lead collimator, the X-ray photons are captured by the PSC X-ray detector. The 3 × 3 detector is equivalent to a single-point macro-pixel in-X-ray-detector computing device, which can obtain images by continuously detecting moving objects.

The center pixel of the detector is subjected to a higher forward bias voltage. If only the signal detected by the center pixel is read out, it

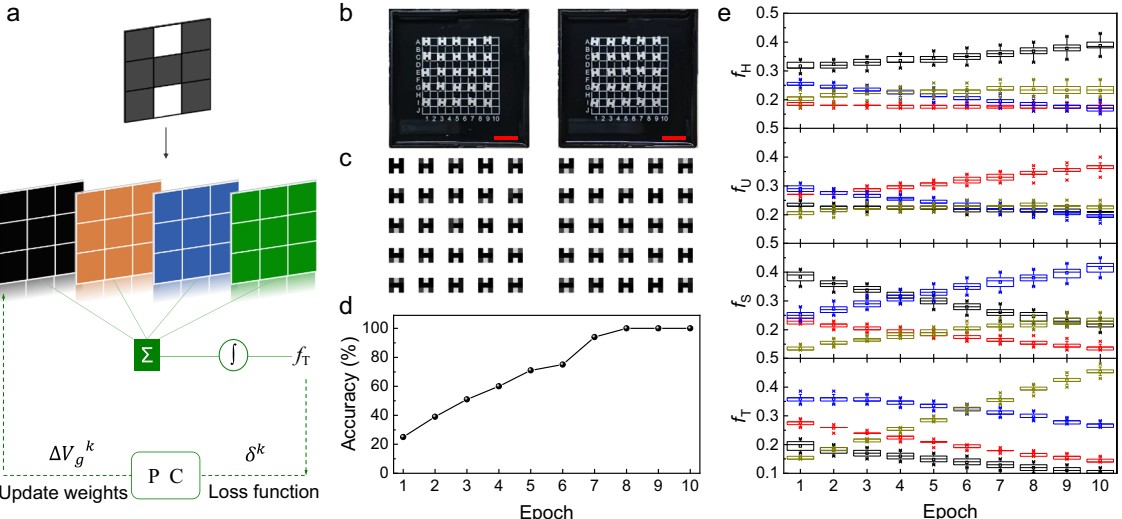

**Fig. 5 | Implementation of a convolutional neural network. a** Training process diagram. PC, personal computer. **b**, **c** Some objects and their detected X-ray pictures are used in the database. All scale bars in red color are 1 cm. **d** Accuracy of recognition over the epochs. **e** Output values, $f_H$, $f_U$, $f_S$, and $f_T$, for each epoch. The four boxplots, the y-axis of which are $f_H$, $f_U$, $f_S$, and $f_T$, respectively, from top to bottom, represent the processing results of the four feature convolution kernels.

The grid lines of the box are divided according to 25%, 50%, and 75% of the data. Percentiles are represented by horizontal lines for minimum and maximum values, crosses for 1% and 99%, and squares for mean values. The colors black, red, blue, and yellow represent the letters H, U, S, and T, respectively. Source data are provided as a Source Data file.

will be the same as the traditional X-ray imaging method, and the result is shown in Fig. 4b (some ragged boundaries may occur due to the translation of steel reticle objects). At this point, reverse bias voltages are further applied to the surrounding pixels, guided by the Laplacian kernel. The final picture in Fig. 4c shows the effect of edge extraction. We can see that, except for the pixels near the edges of the objects, the gray values of other pixels are closer to the background. The macro-pixel detector can achieve the effect of the processing of the Laplacian convolution kernel without software-based computing. Further analysis shows that the pixels with grayscale values of less than 2 constitute 53.6% (4820/8990) of the total, as the grayscale histogram of the image is shown in Fig. 4d. If a reconfigurable integrated detector is used and the threshold comparison is integrated into the CMOS array circuit, the purpose of only transmitting the electrical signals of the effective pixels at the edge will be easily achieved. It can significantly reduce the amount of data, achieving a compression ratio of 46.4% (100-53.6 = 46.4%).

The macro-pixel PSC detector shows the functionality of simultaneous sensing and processing, and the target object can only be detected and transmitted edge information. To further illustrate the effectiveness of this method for data compression, then, we perform computer simulations using massive images from publicly available databases. The image Baggages-0002-0001 in Fig. 1a, from the GDXray + database[50], is elected as a representative and shown again in Fig. 4 for illustration. This public database, GDXray+, contains 20,966 X-ray images in categories such as castings, welds, baggage, and settings. Figure 4e contrasts the original grayscale histogram with the grayscale histogram after image stylization[12,51]. As the simulation result shows, the latter can achieve a data compression ratio of 46.6% by not transmitting pixel signals with gray values less than 1. The average compression ratio of all 20,966 images in the GDXray+ database is 58.8%, which is similar to our experimental results. This indicates that the in-X-ray-detector computing imaging, based on the Laplacian convolution kernel, will promise to achieve about a data compression of ~50% in practice.

Notably, the bias voltages applied to the macro-pixel PSC detector are reconfigurable. It means that we can re-apply differing bias voltages on multiple pixels to implement kernels with other functions, such as the edge softening or noise mean filtering effect in Fig. 4f, the

horizontal and vertical Sobel operator processing in Fig. 4g, h, and one kind of Kirsh processing in Fig. 4i. On the basis of the above, we implement the heterogeneous integration of PSC according to the process flow in Supplementary Figs. 8, 9. Supplementary Fig. 10a shows the blade edge image of the integrated detector using traditional methods. Supplementary Fig. 10b–e shows the simulation effects of various convolution kernels above. Since the placement of the blade edge is directional, it is obvious that the y-direction Sobel kernel has the best edge extraction effect. In addition, for simple objects such as the edge of a blade, their image can achieve better data compression rates. Supplementary Fig. 10b can obtain a data compression ratio as, 275/1024 = 26.8% at a threshold of 10% of maximum brightness, or 418/1024 = 40.8% at a threshold of 5%.

## Implementation of a convolutional neural network

The macro-pixel PSC X-ray detector, with its adjustable weights, shows potential in forming a convolutional neural network and performing image classification tasks[11,14,52], as shown in Fig. 5a. By adjusting the bias voltages across each pixel, it's possible to update the weights of the network. The X-ray information, obstructed by the object and represented as electrical signals, is multiplied by the weight values signified by the bias voltages to compute the total output current. Its output value feedback modulates the bias voltage of each pixel, and the process is repeated.

Real objects are sampled to create a new dataset used to train the network. (The equivalent 3 × 3 convolution kernel of this weight value is [0 0 0, 0 1 0, 0 0 0]) In practical application, we use iron sheets shaped into the letters H, U, S, and T at varying horizontal displacements and rotation angles for testing, with a size of 3 × 3 mm² as depicted in Fig. 5b. And they are detected by the discrete PSC detectors above and sampled as images in the dataset. This makes the gray values of the image in Fig. 5c corresponding to each H letter iron sheet different after sampling. The same operation is applied to the letters U, S, and T, as shown in Supplementary Fig. 12. Initially, the weights are randomly assigned for processing images in the training dataset. The network contains the convolution layer and the softmax activation layer. During each epoch, the network outputs 4 probability values, with a sum of 1, through the activation layer, representing the probability of the image being identified as one of the letters, such as $f_H$ for

the letter H and $f_T$ for the letter T. The largest probability among them is considered the epoch's output, and the accuracy is defined as the correct identifications' proportion for all images in the test dataset. Loss Function is defined by Cross Entropy, from which we perform gradient optimization and get new weight values. The convergence of neural network outputs is assessed by examining the precision of image recognition over various epochs, subsequent to dividing the dataset, conducting training, and performing tests. As shown in Fig. 5d, the accuracy on the test dataset reaches 100% within less than 8 epochs. Other computational procedures with different initial values are shown in Supplementary Figs. 13 and 14, which also shows that the model is convergent. Figure 5e presents the detailed evolution process of the training dataset. Among them, the four sub-figures from top to bottom represent the processing results of the four feature convolution kernels, and the ordinate is the output of each convolution kernel ($f_H, f_U, f_S$, and $f_T$). The black, red, blue, and yellow boxplots represent the data distribution characteristics of numerous image results of H, U, S, and T letters under a certain convolution kernel, respectively. More details about the convolutional neural networks can be found in Methods, Dataset, and Code. As the epochs increase, the extremum and quartile values of each boxplot with different colors are separated in turn, which proves that the model achieves the task of classifying the target images.

This experiment demonstrates that the reconfigurable-weight macro-pixel PSC X-ray detector can effectively identify objects in a specific scene without the help of software algorithms (i.e., distinguishing between iron sheets shaped as H, U, S, and T) and is easily compatible with random placement errors (less than ±15°, 1/5 macro-pixel size, according to Fig. 5b). The function holds potential applications in fields dealing with complex and multi-solution problems in reality, such as the identification of dangerous objects in security inspection, or the detection of tiny nodules in medical inspection, and so on.

## Discussion

In this article, we report the in-X-ray-detector computing device. Benefiting from the low defect concentration and effective passivation, the N-I-P type $CsPbBr_3$ PSC detector has achieved ideal performance for in-sensor computing, including polarity reconfigurability, good linear dynamic range, and robust stability. The hardware itself can perform edge extraction for the data compression, process images using distinct kernels, and perform pattern recognition tasks with high accuracy, which is equivalent to the performance of complicated and energy-extensive software-based computing processes. This research offers potential solutions to overcome challenges associated with data transmission, processing, and storage in X-ray detection systems, paving the way for future development and application of advanced neural network-based X-ray detectors.

## Methods

### Materials

CsBr (99.999%), $PbBr_2$ (99.999%), and $C_{60}$ (99.9%) were from Advanced Election Technology Co. Ltd. And were further purified. The quartz ampoules were from KaiDe (Beijing) quartz Co., Ltd. Abrasives and tools were from Trojan (Suzhou) and Kejing (Shenyang) Co. Ltd. The ZnO target (99.99%), NiO target (99.9%), Au particles (99.999%), and Bi particles (99.999%) were from ZhongNuo Advanced Material (Beijing) Co. Ltd.

### Preparation of $CsPbBr_3$ single crystal

We mixed CsBr and $PbBr_2$ in the quartz ampoule according to the molar ratio, sealed the ampoule, and placed it in a swing furnace to fully react. The composite $CsPbBr_3$ polycrystalline raw material was first purified by directional solidification and then subjected to several zone-melting passes at variable speeds and temperature fields. After

removing the head and tail impurity-rich regions, the ingot had a purity of 99.9999% (6 N, GDMS tested by www.sci-go.com, as shown in Supplementary Table 3) and was used to grow single crystals. The $CsPbBr_3$ perovskite single crystals were grown using the Bridgman quartz-crucible descent method, through a temperature gradient of 1.05 K mm$^{-1}$ at a velocity of 0.40 mm h$^{-1}$. The ingots that were formed were then gradually cooled to ambient temperature at rates of 8 K h$^{-1}$ (initial phase) and 1 K h$^{-1}$ (during phase transition stages). Then, the crystal went through several process steps, including sample mounting, ingot cutting, various mesh-abrasive-paper grinding, and ethanol-DMSO-mixed-solution polishing.

### Device fabrication

We successively conducted thermal evaporation of the $C_{60}$ (~6 nm, through the ellipsometer analysis) layers on both sides of the $CsPbBr_3$ PSC, magnetron sputtering of ZnO (~200 nm) and $NiO_x$ (~150 nm), and thermal evaporation of bismuth electrode layer (~4 μm). Aluminum-cored carbon-side tape (7311, Nisshin EM) was used to peel off the functional layers. More information about the detector can be found in Supplementary Figs. 2 and 3. We showed the discrete, reconfigurable PSC macro-pixel detector in Supplementary Fig. 11.

### Detector performance measurement

Different bias voltages were applied and signals were collected using the Keithley 6517B Source Meter. To assess the X-ray detection capabilities, the Amptek Mini-X2 tube (Au target, Newton Scientific M237) was employed to generate X-ray photons, and a horizontal displacement stage was utilized for moving the object along the $x–y$-axis. And the accu-diode ionization chambers included DDX6-W and DDX6-WL (Accu-Gold+, Radcal). During the test process, we used the photocurrent to subtract the dark current and fit the $I–V$ curve with the Hecht formula.

### Processing and testing of the integrated detector

The CMOS dice were taped out at Semiconductor Manufacturing International (Shanghai) Corporation, using the 180 nm 1P6M Mixed Signal process. On the CMOS wafer, we first performed under-bump-metal plating and then performed dicing and gold wire ball bonding at Shandong Senspil Semiconductor Co., Ltd. After bonding the CMOS die and the self-made printed circuit board (PCB), we used the 6005 T anisotropic conductive tape for the flip-chip bonding, via the bonder, FC150. During imaging tests, the control signals were input by FPGA. We used the oscilloscope (Keysight DSO-S054A) to get the serial output differential signals, which were processed and composed of images in the software. We demonstrated the experiments based on the PSC-CMOS detector in Supplementary Figs. 8–10.

### Classification task

We used the discrete PSC detector to acquire images from which the database was formed. For the classification task, our dataset contained 200 images. The iron pieces corresponding to each letter were photographed 50 times in different placement states. They were divided into two parts, with 160 images in the training set and 40 in the test dataset. We simulated changes in sub-pixel weights through software operations on the computer. The backpropagation process and weight update process referred to the measured signal value of the discrete PSC detector. The results of the training set were output in Fig. 5e, and the results of the test dataset were output in Fig. 5d. The attachment contained Dataset and Code files.

## Data availability

The X-ray images including Baggages-0002-0001 in this study are provided in the public GDXray+ database, under the accession website, https://domingomery.ing.puc.cl/material/gdxray/. On the site, all X-ray images are available at the Download tab. The self-made database for

convolutional neural networks, including all letter images used in Fig. 5, is provided in the attachment Supplementary Data 1. The sub-file names, h, u, s, and t, respectively, represent the corresponding imaging objects. All data is free to use. Source data are provided in this paper.

## Code availability

The source code for convolutional neural networks is available in the folder Supplementary Software 1.

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

## Acknowledgements

This work was supported by the Major State Basic Research Development Program of China (2021YFB3201000 to G.N.), National Natural Science Foundation of China (62134003 to J.T., 62074066 to G.N., 62204092 to H.W.), Fund for the Natural Science Foundation of Hubei Province (2021CFA036 to G.N., 2020CFA034 to J.T.), Shenzhen Basic Research Program (JCYJ20200109115212546 to J.T.), Fundamental Research Funds for the Central Universities (YCJJ202203001 to J.P.) and China Postdoctoral Science Foundation (2022M710054, 2023T160242 to H.W.). The authors would like to thank Youwen Ren for the GDMS and ellipsometer analysis, Professor Wei Xiong for CLSM, and Assistant Qingli Zhang.

## Author contributions

G.N. and J.T. conceived and supervised the project. J.P. performed the growth of the PSC, device fabrication, CMOS integration, and demonstration for various applications. H.W. carried out the characterization of the device and the numerical simulations for the intrinsic sensitivity. H.L. completed the CMOS layout design and PCB production. T.J. did the detector imaging. J.P. and G.N. wrote the paper and all authors commented on the manuscript.

## Competing interests

The authors declare no competing interests.
