## [Peer Review File · Nature Communications]

Reconfigurable perovskite X-ray detector for intelligent imagingREVIEWER COMMENTS

Reviewer #1 (Remarks to the Author):

In this manuscript, Pang and co-authors proposed a CsPbBr₃-based perovskite X-ray detector, of which a 3x3 macro-pixel can be used to perform in-sensor processing, e.g., edge detection, by applying varying voltages to individual sub-pixel devices. Notably, the authors enhanced the linearity of this photodetector by introducing a C60 protective layer on the perovskite material, eliminating the surface defect and thereby achieving improved response linearity. However, while this paper presented its proof-of-concept through single-pixel measurements and subsequent computational simulations, I believe it falls short in terms of completeness for publication in Nature Communications. Specifically, demonstrating the practical utility of the 3x3 device array for in-sensor processing would significantly enhance the paper's impact. To fulfill the publication criteria, I recommend a major revision before its publications. The following comment would helpful to improve the quality of this manuscript.

Comment #1: I strongly request that the authors would consider demonstrating in-sensor processing using the real devices rather than relying on computational simulations. This can be achieved by integrating (or connecting) 3x3 subpixels to create one macro-pixel, applying optimized voltages to each subpixel to form a kernel, and subsequently measuring the current output while exposing the device array to X-ray images. This process could be further improved by utilizing scanning mechanism across the device array or X-ray image, which would be helpful to acquire the images as illustrated in Figure 4b.

Comment #2: Furthermore, to enhance the clarity of this manuscript, I recommend including a schematic or a photograph explaining the array structure. These should include details about the measurement method used for applying different voltages to the subpixels. In particular, regarding the explanation of Kirchhoff's law ($\sum jR_j$), it appears somewhat ambiguous in its current form. Therefore, it is recommended to provide additional clarification on this matter.

Comment #3 : In the manuscript, the introduction of an additional C60 layer as an effective passivator is discussed, but this assertion lacks substantiating references. For more comprehensive understanding, the authors need to incorporate relevant references that showcase the proven efficacy of C60 layer as passivation materials. Furthermore, there is an ambiguity regarding the presence of the C60 layer between the CsPbBr₃ and NiOX layers. To address this, it is recommended to include the specific fabrication procedure and design in the Methods section..

Comment #4 : The authors discussed the linear responses and the on-off responses under both forward and reverse bias conditions, as shown in Figure 2e and 3c. However, the manuscript lacks a description on the specific methodology concerning the application of voltage inputs and X-ray inputs at each distinct time during the measurement. For better understanding, it is recommended to include detailed information on how those inputs were applied for each measurement instance.

Comment #5 : Considering that the compression ratio can be influenced by the original image, it's important to understand the variability in the reported average compression ratio of 58.8% for the GDXray+ database. What is the standard deviation value for the average compression ratio? Additionally, is there a specific reason for selecting the image Baggages-

0002-0001 as an example throughout the manuscript?

Comment #6 : The legend in Figure 1b is difficult to read. I recommend that the author either improves the legibility of the legend or includes an explanatory statement.

Comment #7 : There are minor mistakes in the manuscript.

- (Page 8, caption of Figure 3) ((e) -> (f))
- (Page 11, line 28) (CDXray -> GDXray)

Reviewer #2 (Remarks to the Author):

The author proposes an intelligent X-ray detector based on CsPbBr₃ single crystals in this article. Through passivation and device design, the detector exhibits excellent performance. The author successfully connects the detector with the CMOS chip and demonstrates edge extraction and data compression imaging using this principle through board-level circuits, showing strong innovation. This work is solid and aligns with the forefront of industry research. I believe it will inspire other researchers in the field, and the article will be suitable for publication in Nature Communications after addressing the following issues.

The following questions are critical and relate to the core innovations of the article.

1. This macro-pixel scheme has a very critical shortcoming. Providing information for the same pixel of the image with sub-pixels will inevitably sacrifice the spatial resolution of the array detector. The author does not mention this issue and its solution in the paper.
2. The paper clearly explains the principle of the proposed scheme at the beginning, but there is less description of the implementation of "software hardening" in the latter part. It is the core innovation of this paper. Especially, as a new and promising material, there have been no reports on the electrical connection between perovskite and array CMOS. This engineering innovation is achieved in this paper. I suggest the author supplement the following explanatory information for reference by other researchers in the field, including how to connect the X-ray detector with the chip electrodes, how CMOS processes pixel signals, and how the system reads and outputs information. (There is insufficient information in Figure 3f and Figure S6.)
3. The author states that "Cd(Zn)Te suffers from poor hole collection capability." As far as I know, CZT is a well-established commercial single-crystal X-ray detector. Why does the author claim that CZT is not suitable for in-sensor computing? And why is the CsPbBr₃ semiconductor, suitable? What about other semiconductors like Si?
4. The author states, "Its output value feedback modulates the bias voltage of each pixel, and the process is repeated." How does the detector use this feedback modulation for target recognition? Is it implemented through a neuromorphic chip in hardware?

There are also some other descriptions where it is recommended that the author rephrase the wording to facilitate better understanding by readers.

1. In the paper, what is the amount of data used for the neural network in Figure 5? Are the images of the letter "HUST" in the training set and test set the same? From my understanding, the data volume was not large, so is there a phenomenon of overfitting? It is suggested to supplement the processing and relevant explanations of the neural network in the Methods section.
2. How was the conclusion reached that "it is easily compatible with random placement errors (less than $\pm 15^\circ$, 1/5 pixel size)"?
3. The author states, "we purified the raw materials by zone-melting multiple times to a purity of around 99.9999%." What does it mean to achieve this level of purity? Readers who work

with detectors may not fully understand this. Please provide an explanation and corresponding evidence.

4. In this work, the author has achieved a detector with an excellent linear dynamic range, which has an advantage compared to other commercial products and published reports. I am curious why the linear dynamic range of X-ray detectors is generally low, while achieving 106 dB for visible light detectors is not particularly difficult. I hope to receive an explanation. In addition, I understand that a larger linear dynamic range is better for all detectors. Does an extremely high linear dynamic range have any additional significance for in-sensor computing?

5. The article is quite professional and covers multiple fields, but it uses many schematic diagrams, which may be difficult for readers without a background in the field to understand. It is recommended to supplement optical photos at key points, such as single crystals, devices, results of backside bonding, and so on.

Reviewer #3 (Remarks to the Author):

The authors reported the CsPbBr₃ single crystal-based X-ray detectors and their in-sensor computing applications. They argued that their device achieved polarity reconfigurability, high linear dynamic range, and robust stability due to the elimination of surface defects in their PSC single crystal by C60 passivation. In addition, they argued that in-sensor computing was demonstrated using with distinct kernels.

Although some of the results are interesting, the manuscript overall lacks sufficient novelty and requires supplementary data to support claims. Some specific technical comments:

1. Please cite the article that the first large-scale demonstration of halide perovskite X-ray imaging: <https://doi.org/10.1038/nature24032>

2. The authors claimed that this result is the first in-X-ray-detector computing with halide perovskite single crystals. They claimed that their linear dynamic range is up to 106 dB, which is the highest level compared with other materials, even perovskites result. However, similar concepts of reconfigurable perovskite single crystal x-ray detectors and even in-sensor x-ray computing in the array-level demonstration have been already reported. (doi.org/10.1109/IEDM45625.2022.10019447) In addition, more than 158 dB of linear dynamic range with perovskite nanocrystals was also reported (doi.org/10.1016/j.isci.2021.102927). Moreover, C60 passivated perovskite single crystal X-ray detectors have also been reported recently (<https://doi.org/10.1038/s41566-023-01207-y>). What is the unique achievement of the result in the perovskite X-ray detector research field compared with previous reports?

3. In Figure 1, does the PSC X-Ray Detector have advantages compared to other commercial products in terms of power consumption?

4. In Figure 3c, what is the y-axis scale? If they are photocurrents, which electrode is grounded? And, how large are their photocurrents?

5. Are there other material characterization data for C60 films? If it has low defects between C60 and PSC, please add some AFM or high-resolution SEM images to verify their surface status.

6. In Figures 3f, 4b, 4c, and 5b, please attach the scale bars in all-optical images and data to explain how large your PSC detectors are, the array size, and the steel reticle objects ("HUST"). How large the X-ray beam size illuminate the array? How can you align the exact point of X-ray to the without monitoring from optical microscopes? Are there any effects or noise of misaligning issues? To help the reader's understanding, please attach the optical

images of CPB single crystal, ingots, and measurement setup.

7. In Fig 5a, how many pixel devices were used for one unit of 3x3 letter images? The author argued that this result is an in-sensor computing demonstration. However, according to their explanation, they did not update their device synaptic weight, just used their PSC array for collecting the X-ray pixel images. Their accuracy result was not from in-sensor computing but from the trained CNN model in the Von-Neumann external processor-based simulation. Even though their model did not consider their in-sensor process, this simulation result is just from the CMOS processor-based CNN learning with 3x3 pixel images.

8. In Fig 5, how many datasets are for the training, and how many datasets are for the test? 100% accuracy is just an overfitted result which means their dataset has a pre-bias before testing. The author mentioned that the details of CNN can be found in Methods, but their Methods did not contain any relevant information, not even the supplementary information.

Reviewer #1 (Remarks to the Author):

In this manuscript, Pang and co-authors proposed a CsPbBr₃-based perovskite X-ray detector, of which a 3x3 macro-pixel can be used to perform in-sensor processing, e.g., edge detection, by applying varying voltages to individual sub-pixel devices. Notably, the authors enhanced the linearity of this photodetector by introducing a C60 protective layer on the perovskite material, eliminating the surface defect and thereby achieving improved response linearity. However, while this paper presented its proof-of-concept through single-pixel measurements and subsequent computational simulations, I believe it falls short in terms of completeness for publication in Nature Communications. Specifically, demonstrating the practical utility of the 3x3 device array for in-sensor processing would significantly enhance the paper's impact. To fulfill the publication criteria, I recommend a major revision before its publications. The following comment would helpful to improve the quality of this manuscript.

Comment #1: I strongly request that the authors would consider demonstrating in-sensor processing using the real devices rather than relying on computational simulations. This can be achieved by integrating (or connecting) 3x3 subpixels to create one macro-pixel, applying optimized voltages to each subpixel to form a kernel, and subsequently measuring the current output while exposing the device array to X-ray images. This process could be further improved by utilizing scanning mechanism across the device array or X-ray image, which would be helpful to acquire the images as illustrated in Figure 4b.

Reply: Thank you for your suggestions. In fact, we employed a real device with 3×3 sub-pixels and scanned the object for imaging in the article, just as the reviewer suggested. **Fig. 4a** is a schematic diagram of the actual imaging system, and **Fig. 4b~c** present the actual imaging results with Laplacian kernels. We have revised the manuscript accordingly to avoid ambiguity.

Additionally, we have added the photos of the real imaging setup (**Figure R1**) in the **Supplementary Information**.

Figure R1. The photos of imaging setup for Fig. 4. (a) Optical photo of the object. (b) Photo of the collimator. (c) Photo of the detector. There are 3×3 sub-pixel

electrodes on the CsPbBr₃ PSC, with an electrode size of 0.8 mm and a space of 0.2 mm. The top sub-pixel electrodes are connected to the printed circuit board and are applied with different bias voltages through the electrometer. The bottom common electrode collects the summed signal for the macro pixel. To achieve the Laplacian kernel effect, we connect the 8 surrounding sub-pixels on the back of the PCB by silver paste and apply the same reverse bias voltage. In contrast, the central sub-pixel was applied a forward voltage. This design facilitates the implementation of edge extraction imaging or other effects in board-level circuits. (d) Confocal laser scanning microscope (CLSM) photo of the sub-pixels. (e) Photos of the test system. A displacement stage moves the object for imaging.

Comment #2: Furthermore, to enhance the clarity of this manuscript, I recommend including a schematic or a photograph explaining the array structure. These should include details about the measurement method used for applying different voltages to the subpixels. In particular, regarding the explanation of Kirchhoff's law ($\sum jR_j$), it appears somewhat ambiguous in its current form. Therefore, it is recommended to provide additional clarification on this matter.

Reply: Thank you for your question. We have provided photographs in the response of Comment #1, and revised the manuscript. Specifically, the carbon electrode serves as the bottom common electrode for connecting to the collection signal probe of the electrometer 6517B. The bismuth electrodes serve as sub-pixel electrodes, and they are connected to the Printed-circuit board passing through 20 μm gold wires by ball bonding, where the solder joints are reinforced with silver paste. The sub-pixel electrodes are applied with different bias voltages through the electrometer.

Furthermore, Kirchhoff's current law states that at any node in a circuit, the algebraic sum of the currents is zero. Here, the electrometer's collection probe collects the signal current of the carbon electrode. "Mathematically, its value is the sum of the currents of each sub-pixel, with the opposite vector of the sum." In intelligent detection devices, such as in-sensor computing, Kirchhoff's current law is widely used as the theoretical cornerstone. (doi.org/10.1038/d41586-020-00592-6 (*Nature* 2020), doi.org/10.1038/s41563-023-01676-0 (*Nature Materials* 2023), doi.org/10.1038/s41928-020-00501-9 (*Nature Electronics* 2020))

Comment #3 : In the manuscript, the introduction of an additional C₆₀ layer as an effective passivator is discussed, but this assertion lacks substantiating references. For more comprehensive understanding, the authors need to incorporate relevant references that showcase the proven efficacy of C₆₀ layer as passivation materials. Furthermore, there is an ambiguity regarding the presence of the C₆₀ layer between the CsPbBr₃ and NiO_x layers. To address this, it is recommended to include the specific fabrication procedure and design in the Methods section.

Reply: Thank you for your suggestion. We have inserted previously reported work, doi.org/10.1021/acsnano.7b08561, doi.org/10.1016/j.electacta.2022.141215, doi.org/10.1021/acs.jpcclett.5b00902, doi.org/10.1016/j.electacta.2018.09.004 in the manuscript as references No. 49-52. These researches studied the passivation effect of

C₆₀ on perovskite devices.

For the presence of C₆₀ between CsPbBr₃ and NiO_x, we want to make it clear that here the thin C₆₀ layer just serves as a passivator rather than a transport layer, since magnetron sputtering would negatively influence the surface of the crystals. “The presence of undercoordinated Pb defects was probably due to surface damage from the magnetron sputtering or polishing process and thus halide vacancies.” We have added content to **Methods**. “We successively conducted thermal evaporation of the C₆₀ (~6 nm) layers on both sides of the CsPbBr₃ PSC, magnetron sputtering of ZnO and NiO_x, thermal evaporation of bismuth electrode layer and Au layer.”

Comment #4: The authors discussed the linear responses and the on-off responses under both forward and reverse bias conditions, as shown in Figure 2e and 3c. However, the manuscript lacks a description on the specific methodology concerning the application of voltage inputs and X-ray inputs at each distinct time during the measurement. For better understanding, it is recommended to include detailed information on how those inputs were applied for each measurement instance.

Reply: Thanks to the author for the suggestion. We have added information to illustrate this issue in the manuscript and **Note IV**.

In the experiment, we first measured the I-V and I-t characteristics. The two devices in **Fig. 2e** are biased at -100V and the change in the X-ray photo-response current is due to the artificial adjustment of the working current of the tube, from 20 μ A to 140 μ A. “The detector displayed a good linear response to the dose rates between 32 to 494 μ Gy s⁻¹.” The linear dynamic range test in **Fig. 3d** also used -100 V. The change in X-ray intensity is achieved by changing the working distance, and fixing the parameters of the tube.

Besides, the device with C₆₀ in **Fig. 3c** is biased at -73V and 95V, respectively. According to the mathematical definition of the Laplacian kernel, the photon-response current should be -1:8 between the surrounding sub-pixel and the central sub-pixels. We made slight adjustments based on the I-V curve in **Fig. 3b**, and the I-t curve in **Fig. 3c**, resulting in the photo-response current ratio of -1:8. It was then applied to the imaging experiment in **Figure R1**.

Comment #5 : Considering that the compression ratio can be influenced by the original image, it's important to understand the variability in the reported average compression ratio of 58.8% for the GDXray+ database. What is the standard deviation value for the average compression ratio? Additionally, is there a specific reason for selecting the image Baggages-0002-0001 as an example throughout the manuscript?

Reply: Thanks for your question. For different types of images, the compression ratio of common lossless compression algorithms is usually between 70% (worse effect) and 20% (better effect). Images with a large number of repeated pixels generally achieve higher compression ratios. Images with rich details and color variation, like the image **Baggages-0002-0001**, are likely to receive lower compression ratios. In this database, it should be noted that due to the different types of imaging objects in the

database, the standard deviation of all data is approximately 13.4% (unit of compression ratio).

In the manuscript, we chose this image because it has rich image details, a wide distribution of grayscale values, and the target object is close to life and has practical significance.

Comment #6 : The legend in Figure 1b is difficult to read. I recommend that the author either improves the legibility of the legend or includes an explanatory statement.

Reply: Thank you for your suggestion. We have modified the legend so that its font is clearer, as shown in **Figure R2**. Besides, in order to maintain concise writing in the main text, we have only presented the conclusions and the intended message of the authors in the manuscript. Correspondingly, we have newly added extensive descriptions in **Note I**, explaining in detail how the data was obtained and the meaning behind the points and lines.

Figure R2. Working mechanism of in-X-ray-detector computing.

Comment #7 : There are minor mistakes in the manuscript.

- (Page 8, caption of Figure 3) ((e) -> (f))

- (Page 11, line 28) (CDXray -> GDXray)

Reply: Thank you for your suggestion. We have made the corrections to these errors and also carefully checked other typos.

Reviewer #2 (Remarks to the Author):

The author proposes an intelligent X-ray detector based on CsPbBr₃ single crystals in this article. Through passivation and device design, the detector exhibits excellent performance. The author successfully connects the detector with the CMOS chip and demonstrates edge extraction and data compression imaging using this principle through board-level circuits, showing strong innovation. This work is solid and aligns with the forefront of industry research. I believe it will inspire other researchers in the field, and the article will be suitable for publication in Nature Communications after addressing the following issues.

The following questions are critical and relate to the core innovations of the article.

1. This macro-pixel scheme has a very critical shortcoming. Providing information for the same pixel of the image with sub-pixels will inevitably sacrifice the spatial resolution of the array detector. The author does not mention this issue and its solution in the paper.

Reply: Thank you for raising this important question. It is right that the macro-pixel scheme is faced with the sacrifice in the spatial resolution. In our study, we used a CMOS detector with a pixel pitch of 46 μm . If these pixels are used to form macro-pixels, the macro-pixel pitch would be 138 μm , which is comparable to mainstream flat-panel detectors (~ 150 μm). Then we think it is still possible to achieve the required resolution for applications while adding in-sensor computing capability, by utilizing a small sub-pixel pitch size.

2. The paper clearly explains the principle of the proposed scheme at the beginning, but there is less description of the implementation of "software hardening" in the latter part. It is the core innovation of this paper. Especially, as a new and promising material, there have been no reports on the electrical connection between perovskite and array CMOS. This engineering innovation is achieved in this paper. I suggest the author supplement the following explanatory information for reference by other researchers in the field, including how to connect the X-ray detector with the chip electrodes, how CMOS processes pixel signals, and how the system reads and outputs information. (There is insufficient information in Figure 3f and Figure S6.)

Reply: Thanks for the good advice. In the revised manuscript, we have added a large number of descriptions, including **Figure R3** and **R4**. They consist of many pictures and descriptions, which complement the content that you mentioned.

Figure R3. Schematic diagram of the heterogeneous integration process. (a) The above figure shows the chemical mechanical polishing process of the PSC. The figure below shows the process of vacuum method preparation of pixelated electrodes using the ultrathin fine mask. (b) After tape-out, golden under-bump metals (UBM) need to be electroplated on the silicon wafer. The figure below shows the circuit diagram of a single pixel. The design ideas can be found in our previous work (DOI: 10.1109/ICSICT49897.2020.9278370). (c) Structure of sub-PCB. The bottom of the CMOS die is fixed on the sub-PCB, golden wire ball bonding is used to draw out the electrical signals of the CMOS, and encapsulation resin is used to protect wires. (d) Schematic diagram of the flip-chip bonding process. First, a bilateral microscope is used to align the pixelated electrodes on the PSC and CMOS, and then anisotropic conductive tape (ACF) is added to realize the electrical connection. The UBM can enter the resin, squeeze the Au/Ni conductive particles, and conduct pixel electrodes up and down under pressure. The resin is solidified by slowly heating it while applying pressure. (e) Schematic diagram of the preparation of the common electrode and the protective electrode on the CsPbBr₃ PSC. (d) Mother PCB is connected to sub-PCB through pin headers. The test process is controlled by Field Programmable Gate Array (FPGA).

Figure R6. Pictures of the heterogeneous integration process and result. (a) The pins of the CMOS die are connected to the lead-out electrodes on the sub-PCB through 20 μm golden wires. This process is affected by sample height and must precede the flip-chip bonding process. Since the gold wire is very fragile, it needs to be protected by resin encapsulation. (b) Connecting the PSC and CMOS using ACF flip-chip bonding. (c) The illustration shows dead pixels, and you can see the missing pixel electrodes on the PSC, the Au/Ni particles on the focal plane, and the aligned UBM underneath. The optical photo without dead pixels is shown in Figure c. (d) The optical image of the mother PCB. The CMOS signals were read out serially.

3. The author states that "Cd(Zn)Te suffers from poor hole collection capability." As far as I know, CZT is a well-established commercial single-crystal X-ray detector. Why does the author claim that CZT is not suitable for in-sensor computing? And why is the CsPbBr₃ semiconductor, suitable? What about other semiconductors like Si?

Reply: Thank you for the question. Now, we have newly added information in the manuscript to elucidate this issue. In simple terms, traditional semiconductor detectors only require a unidirectional bias voltage, which places requirements on one type of charge carrier. In contrast, the in-X-ray-detector computing discussed in the article requires the simultaneous application of forward and reverse bias voltages. However, "Cd(Zn)Te suffers from poor hole collection capability and limited dynamic range at reverse bias." Perovskite materials, like CsPbBr₃, have more balanced charge collection capabilities and show large dynamic ranges in both voltage directions (both charge carriers).

Silicon is widely used in soft X-ray scenarios (<20 keV). But it is not suitable for the high-energy X-ray detection used in this article, due to its lower attenuation coefficient.

4. The author states, "Its output value feedback modulates the bias voltage of each pixel, and the process is repeated." How does the detector use this feedback modulation for target recognition? Is it implemented through a neuromorphic chip in

hardware?

Reply: Thank you for your question. We have updated the exposition in the manuscript as below, “we showed a discrete, reconfigurable PSC macro-pixel detector, and the imaging data of it in **Fig. 4-5.**” “We simulate changes in sub-pixel weights through software operations on the computer. The back propagation process and weight update process refer to the measured signal value of the discrete PSC detector.”

There are also some other descriptions where it is recommended that the author rephrase the wording to facilitate better understanding by readers.

1. In the paper, what is the amount of data used for the neural network in Figure 5? Are the images of the letter "HUST" in the training set and test set the same? From my understanding, the data volume was not large, so is there a phenomenon of overfitting? It is suggested to supplement the processing and relevant explanations of the neural network in the Methods section.

Reply: Thank you for your question. For the neural network used in **Fig. 5**, “our dataset contains 200 images. They are divided into two parts, with 160 images in the training set and 40 in the test dataset.” They don't use the same images. For the third question, we can see that the results of the training set are output in **Fig. 5e**, and the results of the test dataset are output in **Fig. 5d**. Overfitting should not perform well on the test set. In contrast, it can be seen that the model performs well on unseen data, so 100% accuracy here is not overfitting. Besides, we have added content about the data collection process, database composition, neural network details, etc. in **Methods**.

2. How was the conclusion reached that "it is easily compatible with random placement errors (less than $\pm 15^\circ$, 1/5 pixel size)"?

Reply: Thank you for your question. We place the target objects and perform image acquisition, as shown in **Fig. 5b**. The collected images are used as data in the training and test dataset. In this process, the target objects have different horizontal offsets and rotation angles. If we can effectively distinguish objects representing different letters in the test dataset, then it can be shown that the detection method is compatible with random placement errors. In this article, the error value is less than $\pm 15^\circ$, 1/5 pixel size.

3. The author states, "we purified the raw materials by zone-melting multiple times to a purity of around 99.9999%." What does it mean to achieve this level of purity? Readers who work with detectors may not fully understand this. Please provide an explanation and corresponding evidence.

Reply: Thank you for your question. In the article, we want to obtain a detector with low defect density, “so it is very important to remove impurity defects in PSC.” We have added the proof in **Table R1**. The purity of this raw material has reached a relatively high level, refer to references like, doi.org/10.1038/s41467-018-04073-3 (~99.999%), 10.1039/d0ja00223b (~99.998%), and doi.org/10.1039/d2tc01679f (~99.998%).

Table R1. Glow discharge mass spectrometry (GDMS) data about the purity of the CsPbBr₃ PSC. The results tabulated with a “<” sign are detection limits. The following table records the concentration information of 72 elements in the sample. The total impurity level determined, excluding the elements outside the detection limit, is about 1 ppm wt. ($\mu\text{g g}^{-1}$). That is, the purity is approximately 99.9999%.

Elements	Concentration (ppm) wt.	Elements	Concentration (ppm) wt.	Elements	Concentration (ppm) wt.
Li	<0.050	Ge	<0.50	Sm	<0.050
Be	<0.050	As	<0.050	Eu	<0.050
B	<0.050	Se	<0.50	Gd	<0.050
F	<0.050	Rb	0.72	Tb	<0.050
Na	0.14	Sr	<0.050	Dy	<0.050
Mg	<0.050	Y	<0.050	Ho	<0.050
Al	0.12	Zr	<0.050	Er	<0.050
Si	0.22	Nb	<0.050	Tm	<0.050
P	<0.050	Mo	<0.050	Yb	<0.050
S	0.08	Ru	<0.050	Lu	<0.050
Cl	<1.0	Rh	<0.050	Hf	<0.050
K	<0.10	Pd	<0.050	Ta	<0.050
Ca	0.18	Ag	<0.50	W	<0.050
Sc	<0.050	Cd	<0.50	Re	<0.050
Ti	<0.050	Sn	<0.50	Os	<0.050
V	<0.050	Sb	<0.10	Ir	<0.050
Cr	<0.050	Te	<0.10	Pt	<0.050
Mn	<0.050	I	0.26	Au	<0.050
Fe	<0.050	Cs	Matrix	Hg	<0.050
Co	<0.050	Ba	<0.050	Tl	<0.050
Ni	<0.050	La	<0.050	Pb	Matrix
Cu	<0.050	Ce	<0.050	Bi	<0.050
Zn	<0.050	Pr	<0.050	Th	<0.050
Ga	0.14	Nd	<0.050	U	<0.050

4. *In this work, the author has achieved a detector with an excellent linear dynamic range, which has an advantage compared to other commercial products and published reports. I am curious why the linear dynamic range of X-ray detectors is generally low, while achieving 106 dB for visible light detectors is not particularly difficult. I hope to receive an explanation. In addition, I understand that a larger linear dynamic range is better for all detectors. Does an extremely high linear dynamic range have any additional significance for in-sensor computing?*

Reply: Thank you for your question. we have elaborated the answer to the two questions in the added **Note V**.

The LDR performance of visible light detectors is typically three to four orders of magnitude (60 to 80 dB) higher than that of X-ray detectors. There are currently no reports using experiments to directly answer the source of this difference. We expect it

comes from the difference in detection principles. Here are several possible reasons.

① X-ray photons can undergo multi-step excitation processes as below, which is much more complicated than the excitation process of visible light. Each step is affected by defects to varying degrees.

Interaction pathway	Products (participating in the cycle)	
	Primary products	Secondary products
Photoelectric effect	Characteristic X-ray	
	Auger electron	Bremsstrahlung and ionization
	Photo-electron	excitation (leading δ -ray)
Compton effect	Characteristic X-ray	
	Scattered photon	
	Compton recoil electron	Bremsstrahlung and ionization
	Auger electron	excitation (leading δ -ray)
Coherent scattering	Scattered photon	
Pair production (requires a high level of energy)		
Photo-nuclear reaction (requires a high level of energy)		

② X-ray detectors have a much greater thickness, making it challenging to perfectly satisfy the condition, $\mu\tau E \gg d$. This can lead to a significant self-limiting response deviation caused by limited carrier mobility at high X-ray intensity.

③ X-ray detectors are prone to radiation-induced damage and defects, leading to trap-limited sublinear response under low X-ray intensity.

For the second question, in-X-ray-detector computing has much higher requirements for linear dynamic range than other detectors. Specifically, what it focuses on, is not the absolute value of the photon-response signal, but the relative difference. Theoretically, under high, medium and low X-ray intensity, the same difference in the number of X-ray photons should show the same signal value on the detector. “In contrast, traditional detectors with low LDR have different sensitivities, when overexposed or underexposed. The photo-current of sub-pixels should have been at the designed ratio, but this ratio changed at this time. The nonlinear response cannot achieve a zero response signal when the sub-pixel signals are summed. Therefore, the effect of the convolution kernel cannot be achieved.”

5. The article is quite professional and covers multiple fields, but it uses many schematic diagrams, which may be difficult for readers without a background in the field to understand. It is recommended to supplement optical photos at key points, such as single crystals, devices, results of backside bonding, and so on.

Reply: Thank you for your suggestion. We have revised the manuscript accordingly. Some of the information is included in **Figure R3-4**. In addition to these above, we have added information, such as **Figure R1** to Reviewer #1, and **Note IV-V** in the **Supplementary Information**, to assist readers in understanding.

Reviewer #3 (Remarks to the Author):

The authors reported the CsPbBr₃ single crystal-based X-ray detectors and their in-sensor computing applications. They argued that their device achieved polarity reconfigurability, high linear dynamic range, and robust stability due to the elimination of surface defects in their PSC single crystal by C60 passivation. In addition, they argued that in-sensor computing was demonstrated using with distinct kernels.

Although some of the results are interesting, the manuscript overall lacks sufficient novelty and requires supplementary data to support claims. Some specific technical comments:

1. Please cite the article that the first large-scale demonstration of halide perovskite X-ray imaging: <https://doi.org/10.1038/nature24032>

Reply: Thank you for your suggestion. We have cited the article as reference No. 43.

2. The authors claimed that this result is the first in-X-ray-detector computing with halide perovskite single crystals. They claimed that their linear dynamic range is up to 106 dB, which is the highest level compared with other materials, even perovskites result. However, similar concepts of reconfigurable perovskite single crystal x-ray detectors and even in-sensor x-ray computing in the array-level demonstration have been already reported. (doi.org/10.1109/IEDM45625.2022.10019447) In addition, more than 158 dB of linear dynamic range with perovskite nanocrystals was also reported (doi.org/10.1016/j.isci.2021.102927). Moreover, C60 passivated perovskite single crystal X-ray detectors have also been reported recently (<https://doi.org/10.1038/s41566-023-01207-y>). What is the unique achievement of the result in the perovskite X-ray detector research field compared with previous reports?

Reply: Thank you for your question.

First, I would like to clarify the issue regarding the linear dynamic range (LDR) of perovskite-based X-ray detectors. In our paper, the tested LDR is specific to X-ray detection. Compared to other reports, our perovskite-based detectors exhibit significant advantages, as demonstrated in **Fig. 3**. The results you mentioned were obtained for visible light detection. In fact, there have been many other studies reporting perovskite-based visible light detectors with LDR performance well above 100 dB (doi.org/10.1002/adma.201703209, doi.org/10.1002/adfm.202306941, doi.org/10.1002/sml.202005626). Additionally, traditional materials like silicon have also been reported to achieve visible light LDRs of over 150 dB. The LDR performance of visible light detectors is typically three to four orders of magnitude (60 to 80 dB) higher than that of X-ray detectors.

Then, the article (doi.org/10.1109/IEDM45625.2022.10019447) implements the in-sensor X-ray computing demonstration of array-level perovskite detectors. But it has many differences from our article.

First, Dun and his collaborators proposed a reconfigurable mechanism based on ion redistribution. Specifically, the distribution of Br⁻ ions can be tuned after a

pooling pulse voltage. The Br element component shows an obvious decrease near the negative electrodes, which leads to the polarity reversal of the built-in electric field in the perovskite layer. Their work and our work are both in-sensor computing. However, the two are based on different principles, and the final effects achieved are also different. Our work is another good complement to the emerging field of in-sensor computing.

In addition, their array-level perovskite X-ray detector and the previous first-large-scale-demonstration work (<https://doi.org/10.1038/nature24032>) are based on the polycrystalline thick film process route. In contrast, our work addresses the issue of process compatibility of perovskite single crystals with circuit arrays. This kind of single-crystal integrated detector can not only be used in the in-sensor computing field but also hopes to be used in other fields in the future. X-ray photon counting imaging is a good example. This is the recognized next-generation technology in the medical field, and in principle, only single crystals can achieve the photon counting detection function, while films cannot. And there are currently no integration reports on perovskite single crystals. We believe that our work has filled the technical gap in perovskite single-crystal integration and will be inspiring in many fields.

Finally, for the passivation by C_{60} , we were indeed inspired by previously published work. But here, through Thermally Stimulated Current (TSC), we propose a result about the passivation principle and point out possible sources of PSC defects. Combining the defect depth measured by TSC and previous theoretical calculation work (doi.org/10.1021/acs.jpcclett.6b02800), we believe that, “the defects were probably from the Pb-Pb dimers, which formed on the surface as a consequence of halide vacancies. After passivation, the device showed negligible defects in the whole range.” This is also the premise to realize such a high LDR X-ray detector for the first time. We believe this will inspire other researchers who are interested in perovskite PSC detectors, perovskite solar cells, and X-ray detectors of other materials.

3. In Figure 1, does the PSC X-Ray Detector have advantages compared to other commercial products in terms of power consumption?

Reply: Thanks for your question. The PSC detector used in this article has lower power consumption compared to other detectors, such as CZT. It is specifically reflected in the following two aspects. On one hand, the calculation that occurs on the detection unit is a parallel process. That is, the detected data are processed at the same time, and the resources consumed by the calculation process are reduced. On the other hand, it reduces the power consumption of frequent data transmission between the detection unit and the calculation unit.

Since the evaluation of power consumption is a systematic project, it involves multiple processes such as the power supply of the detector circuit by the slip ring, the power supply of the data sending and receiving equipment, and the power consumption of the workstation data processing. Generally, a system needs to build a software model first before it can be evaluated rigorously. Below, we assume that a

64-row, 128-slice CT equipment introduces the in-X-ray-detector computing method. Then we make a rough estimate. There is no change in the power supply of its detector (the power of a single chip is tens of watts, usually there are 40 pieces), and the data transmission process is reduced from 2 cycles to 1 cycle (the power of the transmitter, receiver, power amplifier, and signal processing equipment is 100~400W), and the data processing process of the workstation (about a few hundred watts to several kilowatts) is simplified by 20%. It can be very roughly estimated that in the process of converting the detected data into images, about 20% of the power consumption is saved in total.

4. In Figure 3c, what is the y-axis scale? If they are photocurrents, which electrode is grounded? And, how large are their photocurrents?

Reply: Thanks for your question. We have supplemented the information in the manuscript, added **Figure R1** to Reviewer #1, and added **Note IV**.

Fig. 3c shows the dark and photo-current curves. The detector is sequentially irradiated with the X-ray, whose intensity gradually decreases. When the dose rate is $494 \mu\text{Gy s}^{-1}$, under the forward bias voltage, the photocurrent is 143 nA at 95V. And it is -25 nA at a reverse bias voltage of -73 V. There is a large difference between the two, so we do not give an absolute value of the y-axes. During testing, “the common electrode serves as a collection electrode, and bias voltages are applied to the sub-pixel electrodes.”

5. Are there other material characterization data for C_{60} films? If it has low defects between C_{60} and PSC, please add some AFM or high-resolution SEM images to verify their surface status.

Reply: Thank you for your suggestion. We have added **Figure R5** to illustrate the characteristics of the C_{60} film, including the optical photo, SEM photo, AFM photo, and ellipsometry test results for thickness.

Figure R5. Characterization of C_{60} layer. (a) Photograph of CsPbBr_3 single crystal with C_{60} passivation layer. (d) SEM image of the C_{60} layer on the surface of the single crystal. (e) AFM image of C_{60} layer. The roughness of the crystal surface is $R_a = 1.15$ nm, and the roughness of the C_{60} surface is $R_a = 2.84$ nm. The ellipsometer (UVISEL Plus, HORIBA) was used to obtain the C_{60} thickness as about 6 nm.

6. In Figures 3f, 4b, 4c, and 5b, please attach the scale bars in all-optical images and data to explain how large your PSC detectors are, the array size, and the steel reticle objects (“HUST”). How large the X-ray beam size illuminate the array? How can you

align the exact point of Xray to the without monitoring from optical microscopes? Are there any effects or noise of misaligning issues? To help the reader's understanding, please attach the optical images of CPB single crystal, ingots, and measurement setup.

Reply: Thank you. We have added scale bars to the above images and newly added optical images, like **Figure R1** and **R4**. The latter contains other information you mentioned.

In this article, the X-ray beam, illuminating the array detector, is about $5 \times 5 \text{ mm}^2$, which is also the size of the collimator and size of the detector.

Since the images in **Fig. 4** and **Fig. 5** all come from the target object movement of the single macro-pixel detector, photons passing through our collimator do not cause the misaligning issue.

7. In Fig 5a, how many pixel devices were used for one unit of 3x3 letter images? The author argued that this result is an in-sensor computing demonstration. However, according to their explanation, they did not update their device synaptic weight, just used their PSC array for collecting the X-ray pixel images. Their accuracy result was not from in-sensor computing but from the trained CNN model in the Von-Neumann external processor-based simulation. Even though their model did not consider their in-sensor process, this simulation result is just from the CMOS processor-based CNN learning with 3x3 pixel images.

Reply: Thanks for your question. We have updated the description in the manuscript to avoid misleading readers. “We showed a discrete, reconfigurable PSC macro-pixel detector, and the imaging data of it in **Fig. 4-5**.” “There are 3×3 sub-pixel electrodes on the CsPbBr₃ PSC, with a size of 0.8 mm and a space of 0.2 mm.” The size of the letter-shaped iron sheets is $3 \times 3 \text{ mm}^2$, as shown in **Fig. 5b**, which is consistent with the size of the detector in **Figure R1**. Therefore, a letter-shaped object can be detected by a macro-pixel detector.

Besides, we obtain the summed signal current by applying bias voltages to the sub-pixels. This is a hardware-level operation that does not go through the von Neumann architecture. “We simulate changes in sub-pixel weights through software operations on the computer. The back propagation process and weight update process refer to the measured signal value of the discrete PSC detector.” If updated weight bias voltages are applied to the sub-pixels, we expect to achieve the same effect as the simulation. And this kind of treatment method is also a common method currently used in recent reports. (doi.org/10.1038/s41586-020-1942-4 (*Nature*, 2020), DOI: 10.1126/sciadv.aba6173 (*Science Advances*, 2020), doi.org/10.1038/s41928-022-00747-5 (*Nature Electronics*, 2022), DOI: 10.1126/science.aaw5581 (*Science*, 2019)) We believe that more and more efforts will be applied to fully integrated in-sensor computing in the future, such as mapping the back propagation algorithm to on-chip hardware and achieving dynamic adjustment of the weights.

8. In Fig 5, how many datasets are for the training, and how many datasets are for the

test? 100% accuracy is just an overfitted result which means their dataset has a pre-bias before testing. The author mentioned that the details of CNN can be found in Methods, but their Methods did not contain any relevant information, not even the supplementary information.

Reply: Thanks for your question. In the revised manuscript, we added a description of the classification task in the **Methods** section, and added Code and Dataset files in the attachment. Here, “our dataset contains 200 images. The iron pieces corresponding to each letter were photographed 50 times in different placement states. They were divided into two parts, with 160 images in the training set and 40 in the test dataset. The results of the training set are output in **Fig. 5e**, and the results of the test dataset are output in **Fig. 5d**.” Overfitting should not perform well on the test set. In contrast, it can be seen that the model performs well on unseen data, so 100% accuracy here is not overfitting. In addition, the classification task has fewer neural convolutional network layers, the images in the dataset have obvious features and great differences, and the number of elements is also relatively small. So, it is possible to achieve 100% accuracy for this simple task in the manuscript.

REVIEWER COMMENTS

Reviewer #1 (Remarks to the Author):

The authors have addressed my concerns well, and therefore, I recommend the publication of this manuscript in its present form.

Reviewer #2 (Remarks to the Author):

The revision meets all my concerns.

Reviewer #3 (Remarks to the Author):

The authors revised their manuscript of the CsPbBr₃ single crystal-based X-ray detectors for in-sensor computing applications. They argued that their work is the first demonstration of perovskite single crystal-based X-ray detector arrays with CMOS.

However, they still show a lack of novelty for being published in the high-impact journal Nature Communication. Therefore, the manuscript still lacks sufficient novelty for publication in Nature Communications. Some specific comments:

1. As reviewers mentioned in Reviewer#3 Q2, their work has novelty from the first-halide perovskite single crystal X-ray detector arrays. However, I have some doubts about their arguments because there were lots of previous works of perovskite single-crystal X-ray detector arrays, such as MAPbI₃ single-crystal X-ray detector arrays (<https://onlinelibrary.wiley.com/doi/full/10.1002/adma.202103078>), MAPbBr₃ single crystal arrays (<https://onlinelibrary.wiley.com/doi/full/10.1002/pssr.201800380>), CsPbBr₃-nIn single crystal arrays (<https://onlinelibrary.wiley.com/doi/10.1002/adma.202106562>), triple cation perovskite single crystal arrays (<https://onlinelibrary.wiley.com/doi/full/10.1002/adma.202006010>; <https://doi.org/10.1038/s41467-023-36313-6>). Moreover, their devices all used C60 as a passivation layer with an in-depth study.

They also argued that their devices can be integrated into the CMOS as 32 by 32 arrays (Figure 3f), but there are no demonstrations or measurements via CMOS circuits in this manuscript. They only used a 3x3 macro-pixel measurement demonstration. Furthermore, CMOS-integrated perovskite nanocrystal X-ray detector arrays were already reported with easy fabrication methods (<https://onlinelibrary.wiley.com/doi/full/10.1002/adma.201801743>). In addition, the authors only used simulation, not the CMOS-circuit-level processing in-sensor computing. Why don't you use CMOS-circuit-based in-sensor processing with 32 by 32 pixels, not the simulation? The 3 by 3-pixel demonstration is not enough to prove the CMOS-integrated array demonstration.

Therefore, I cannot find the novelty of their work from their manuscript including their material-level, device performances, array-level processing, and simulation works (<https://onlinelibrary.wiley.com/doi/full/10.1002/advs.202205536>; <https://onlinelibrary.wiley.com/doi/full/10.1002/adfm.202210335>). What is the novelty of this study in the PSC X-ray detector research field?

2. What is the response time and decay time of your devices compared to other PSC-based X-ray detectors? What are the dynamics in their interface engineering?

3. Please show a comparison chart of perovskite X-ray detectors with your works including active materials, power consumption, device size, LDR, sensitivity, stability, response time, array size, etc to highlight your work.

4. In Figure S2, If the thickness of C60 is 6 nm, and the roughness is 2.84 nm, then the black spots in Fig. S2c are regarded as pinhole defects, which can be trap sites and degrade the device performances (<https://pubs.acs.org/doi/10.1021/acsenergylett.0c02573>). How can we verify the defect-free device by C60 passivation by those results?

Minor comments:

1. Please add the scale bar in Figures S2a and b. In addition, Please change Fig. S2a to the top-view photo of the CsPbBr₃ single crystal with a C60 passivation layer.
2. In Figure S8, please change the reference form from DOI to reference number.

Reviewer #3 (Remarks to the Author):

The authors revised their manuscript of the CsPbBr₃ single crystal-based X-ray detectors for in-sensor computing applications. They argued that their work is the first demonstration of perovskite single crystal-based X-ray detector arrays with CMOS.

However, they still show a lack of novelty for being published in the high-impact journal Nature Communication. Therefore, the manuscript still lacks sufficient novelty for publication in Nature Communications. Some specific comments:

1. As reviewers mentioned in Reviewer#3 Q2, their work has novelty from the first-halide perovskite single crystal X-ray detector arrays. However, I have some doubts about their arguments because there were lots of previous works of perovskite single-crystal X-ray detector arrays, such as MAPbI₃ single-crystal X-ray detector arrays (<https://onlinelibrary.wiley.com/doi/full/10.1002/adma.202103078>), MAPbBr₃ single crystal arrays (<https://onlinelibrary.wiley.com/doi/full/10.1002/pssr.201800380>), CsPbBr_{3-n}I_n single crystal arrays (<https://onlinelibrary.wiley.com/doi/10.1002/adma.202106562>), triple cation perovskite single crystal arrays (<https://onlinelibrary.wiley.com/doi/full/10.1002/adma.202006010>; <https://doi.org/10.1038/s41467-023-36313-6>).

(1-1) Reply: Thanks for your comment. We apologize for not providing a clear explanation in the previous version of our reply. Allow us to provide a detailed explanation here.

In our manuscript, we mentioned, “As far as we are concerned, there is no report on the heterogeneous integration of PSC on CMOS”. In the previous version of our reply, in response to Review #3 Question 2, we stated, “our work addresses the issue of process compatibility of perovskite single crystals with circuit arrays”. These statements differ in meaning from what you mentioned, “their work has novelty from the first-halide perovskite single crystal X-ray detector arrays”.

To clarify more clearly, one of the innovations in our article is **the demonstration of the CMOS circuit-integrated detector based on perovskite single crystals**. The integrated detector encompasses various architectures as a whole. Specifically, it includes modules such as the integrating capacitor, timing control, digital control logic, correlated double sampling, programmable gain amplifier, and driver. It enables simultaneous transient capture of a single image, high spatial resolution of 4.2 lp mm⁻¹, and dynamic imaging at >30 fps, as depicted in the figure below. The imaging process is fully automated, encompassing signal integration and readout. The overall change in detector architecture allows it to function **without the need for external source meter equipment**. Such capabilities have not been achieved in perovskite single crystal-based detectors before, and they cannot be accomplished by simply concatenating individual point detectors into an array.

In contrast, the articles you listed are all excellent works in the field of PSC, but they are not chip-integrated, like CMOS, TFT, CCD, or APS-integrated detectors. Specifically,

① Article 1 (doi/full/10.1002/adma.202103078) employs a displacement platform to move discrete detectors for acquiring X-ray images. This is evident in The X-Ray Detector Characterization in Experimental Section in their paper, where the Keithley 7001 switch system and Keithley 2400 source meter are utilized.

② Article 2 (doi/full/10.1002/pssr.201800380) implements a PCB-level perovskite single crystal detector, which is not chip-integrated. Readers can see this in Figure S7 of their paper. The figure shows a typical PCB-level detector with the chip debugged and soldered on the PCB (including those with ambiguous model numbers on the left, and that numbered SRV7671 on the right). This PCB-level circuit can only realize the function of selecting the pixels of the array detector, and cannot realize the above-mentioned CMOS functions.

By the way, in our manuscript, the 3×3 PSC in **Figure S7**, which implements the effect demonstration of the Laplacian convolution kernel, is realized on a PCB-level detector.

③ Article 3 (doi/10.1002/adma.202106562) also employs discrete single-point detectors, and the Keithley Source Meter 6500 series is used for testing, as shown in Figure S9a and Figure S10e of their paper.

④ Article 4 (doi/full/10.1002/adma.202006010) utilizes single-point detectors and employs an X-Y displacement platform for imaging, as mentioned on page 8. Moreover, the author emphasized in page 8, that the X-ray imaging measurement in Article 4 is a conceptual demonstration. The detectors need to be integrated to linear detector arrays or large flat panel arrays for fast and high-resolution X-ray imaging. And the PSC X-ray imaging system will be able to realize after integrating the PSCs onto the readout circuitry.

These points were realized in our article, which also verified the innovativeness of our work from the side.

⑤ Article 5 (doi.org/10.1038/s41467-023-36313-6) uses Source Measurement Units (SMU) to test the discrete detector, see X-ray photocurrent measurement in Methods. The SMUs are 3 NI PXIe-4138 and 1 NI PXIe-2531 from Emerson. These are multi-channel source meters, which are equivalent to the Keithley 6517B Source instrument used in our manuscript.

In summary, currently, there are no articles reporting on CMOS-integrated PSC detectors, or any other chip-integrated ones.

Besides, it must be emphasized that the CMOS-integrated PSC detector represents an important milestone in the development of X-ray detection and perovskite fields. On one hand, CMOS-integrated detectors offer smaller sizes and weights, eliminating the need for SMUs. This enables efficient acquisition of high-performance images and meets practical application requirements in confined spaces. It is the necessary pathway for industrializing perovskite single crystals in the X-ray field. On the other hand, integration allows for more complex functionalities, via adding modules to the CMOS, without the need for PSC device optimization. For instance, the inclusion of comparators and counters enables photon counting detection

imaging, recognized as the next-generation CT detection method. Alternatively, by adjusting the voltage of pixels on the CMOS, intelligent detection imaging can be achieved, following the approach described in this paper.

Here, for the first time, we propose a semiconductor process flow that meets PCS's performance requirements, including avoiding phase change at high temperatures, avoiding incompatibility with H₂O-based semiconductor processes, and matching parameters with the integrated circuits of response speed, dark current, and sensitivity. Through imaging results, we demonstrate the feasibility of the ACF-based flip-chip bonding route. This innovation provides valuable insights for the development of all halide perovskite single crystals in both academic and industrial settings.

Moreover, their devices all used C₆₀ as a passivation layer with an in-depth study.

(1-2) Reply: Thanks for your comment.

As we said in the previous response letter, for the passivation by C₆₀, we were indeed inspired by previously published work. But here, through Thermally Stimulated Current (TSC), we propose a result about the passivation principle and point out possible sources of PSC defects. Combining the defect depth measured by TSC and previous theoretical calculation work, we believe that, the defects were probably from the Pb-Pb dimers, which formed on the surface as a consequence of halide vacancies. After passivation, the device showed negligible defects in the whole range. This is also the premise to realize such a high LDR X-ray detector for the first time. We believe this will inspire other researchers who are interested in perovskite PSC detectors, perovskite solar cells, and X-ray detectors of other materials.

Compared to the articles you mentioned, we pointed out the source of the defects, and explained the underlying principles of how C₆₀ works. This is what they didn't report.

They also argued that their devices can be integrated into the CMOS as 32 by 32 arrays (Figure 3f), but there are no demonstrations or measurements via CMOS circuits in this manuscript. They only used a 3x3 macro-pixel measurement demonstration.

(1-3) Reply: Thanks for your comment.

To provide better clarification, we have added images of the demonstrations or measurements via CMOS circuits in **Figure R1**. (That is, Figure S6 in the previous version has been supplemented.) Figures a-c show the processing of PSC, Figures d-h show the processing of CMOS, Figures i-k show the ACF-flip-chip process, and Figures l-o show the measurement and demonstration about the CMOS-integrated PSC detector.

Figure R1. Pictures of the heterogeneous integration process and result. (a) Optical photos of the Bridgman's grown ingot and the PSC samples after cutting and polishing. (b-c) PSC with pixel electrodes on the surface. The size of a single pixel is $46 \times 46 \mu\text{m}^2$, and its electrode size is $30 \times 30 \mu\text{m}^2$, which are all complete, uniform in size, and have little evaporation shadows. (d) The CMOS layout, corresponds to the optical photo in **Fig. 3f**. (e-f) CMOS die with UBM. The UBM is made of gold and has a height of $10 \mu\text{m}$, which is enough to squeeze the conductive particles in the ACF. Alignment male keys are reserved, as shown in the red box. (g-h) The pins of the CMOS die are connected to the lead-out electrodes on the sub-PCB. Since the gold wire is very fragile, it needs to be protected by resin encapsulation. (i-k) Connecting the $370 \mu\text{m}$ thick PSC and CMOS using ACF flip-chip bonding. We can observe whether ACF is connected well in the optical photograph, by looking from above on the PSC without electrodes. The illustration shows dead pixels, the Au/Ni particles, and the aligned UBM underneath. The optical photo without dead pixels is shown in Figure k. (l) Enlarge image of Figure 3f. (m) The electrical signals scanned and outputted in a specific row. (n) The signals under different bias voltages. (o) Photo of the testing system. During the measurement, the CMOS circuit is adjustably biased by a DC power with an oscilloscope observing the signal waveforms at various contact points, and a computer controls the operation of the X-ray tube.

Furthermore, CMOS-integrated perovskite nanocrystal X-ray detector arrays were

already reported with easy fabrication methods
(<https://onlinelibrary.wiley.com/doi/full/10.1002/adma.201801743>).

(1-4) Reply: Thanks for your comment.

The authors in the article (doi/full/10.1002/adma.201801743) used a commercial Si-CMOS integrated detector (remote redevye HR, Teledyne Dalsa), and “placed” scintillators on it. It is based on the indirect detection method via scintillators, which is completely different from the direct detection via semiconductors in our manuscript. The “place” method as described by the authors, cannot make electrical contact with semiconductors and it cannot be used on direct-detection X-ray detectors. It is recognized that, in principle, the direct detection method has higher sensitivity and can also achieve higher spatial resolution.

In addition, the authors only used simulation, not the CMOS-circuit-level processing in-sensor computing. Why don't you use CMOS-circuit-based in-sensor processing with 32 by 32 pixels, not the simulation? The 3 by 3-pixel demonstration is not enough to prove the CMOS-integrated array demonstration.

(1-5) Reply: Thanks for your question. This is an excellent question and goes straight to the core pain points of the fields.

Besides our work, several other in-sensor computing studies have experimentally demonstrated the process via software or external digital processors to implement the algorithms. (10.1038/s41586-020-1942-4, *Nature*; 10.1038/s41928-019-0270-x, *Nat. Electron.*; 10.1038/s41928-019-0221-6, *Nat. Electron.*; 10.1126/science.aaw55, *Science*; 10.1038/nature14441, *Nature*; 10.1038/s42256-018-0001-4, *Nat. Mach. Intell.*) These researchers, including us, want to demonstrate them through a fully integrated detector. However, realizing a complete fully integrated CMOS chip, containing a detector, processor, and backpropagation algorithm, remains challenging. The key challenge lies in the inefficiency of mapping other software algorithms, except convolution kernels, to on-chip hardware, such as applying differential voltages to adjacent pixels, operating high-precision data processing with low consumption during weight update, and achieving efficient parallel conductance tuning with write verification. These are difficult research topics for the entire in-sensor computing field. Therefore, most researchers choose the method of "experimental verification of some functions and computer simulation of the remaining functions". From our perspectives, the full-on-chip CMOS-based detector is an independent innovation from the edge-extraction intelligent X-ray imaging concept and demonstration in the article, and is not the content to be solved.

As far as we know, so far, only a few works have implemented full-chip integrated in-sensor computing (DOI: 10.1126/science.ade3483, 2023, *Science*, titled “Edge learning using a fully integrated neuro-inspired memristor chip”)

As a supplement, we have added imaging results in **Figure R2**. Figure a is the actual CMOS-based X-ray imaging of the slanted-edge, along with its Modulation transfer function (MTF) as shown in Figure f. Through image stylization, we simulated the imaging effects in Figures b-e of a CMOS-based PSC detector with a convolution kernel modulation function. **We demonstrated convolution**

kernel-based intelligent imaging through PCB-level circuits, and demonstrated PSC CMOS-based detector imaging through integration in the article. The simulation here envisions the effects of full-on-chip integrated in-X-ray-detector computing.

Figure R2. The result of imaging in reality, and the simulation results. (a) X-ray imaging of the slanted-edge with the 32×32 CMOS-based detector. (b-e) Simulation results via the image stylization, respectively represent Sobel, LOG, Laplacian, and mean filtering convolution kernels, according to **Figure 3** in the manuscript. (f) The modulation transfer function of a, compared with the work, doi.org/10.1038/s41928-021-00644-3.

Therefore, I cannot find the novelty of their work from their manuscript including their material-level, device performances, array-level processing, and simulation works (<https://onlinelibrary.wiley.com/doi/full/10.1002/advs.202205536>; <https://onlinelibrary.wiley.com/doi/full/10.1002/adfm.202210335>). What is the novelty of this study in the PSC X-ray detector research field?

(1-6) Reply: Thanks for your comment.

The main innovation of this research is that we proposed an **intelligent X-ray imaging method** based on convolution kernels and realized the Laplacian convolution kernel effect in experiments. Our CsPbBr_3 single crystal in-X-ray-detector computing method is different from other works in principle (such as doi.org/10.1109/IEDM45625.2022.10019447). In the demonstration, we can achieve different image processing effects such as edge extraction, denoising, three-dimensionalization and so on, 50% data compression rate, and target object resolution functions. It brings a lot of inspiration to the PSC X-ray detector research field.

Other innovations include,

- ① the demonstration of the CMOS circuit-integrated detector based on PSC;
- ② the attribution of PSC defects and the C_{60} passivation effect;
- ③ and record-high linear dynamic range for X-ray detectors.

2. What is the response time and decay time of your devices compared to other PSC-based X-ray detectors? What are the dynamics in their interface engineering?

Reply: Thanks for your question.

Our CsPbBr₃ PSC has a response time of 2.89 ms and a decay time of 2.50 ms to X-ray as shown in **Figure R3**. And it has a response time of 30 μs and a decay time of 710 μs to 365 nm visible light. We expect that it comes from the difference in detection principles. The comparison results with other reports are listed in **Table R1** and **R2**.

Figure R3. The response and decay time. (a-c) Photo-response to X-ray under different abscissas. (d-f) Photo-response to X-ray. The response time is represented by the time it takes for the signal to rise from 10% to 90%, and the decay time is represented by the time it takes for the signal to fall from 90% to 10%.

Table R1. Response time comparison of PSC X-ray detectors.

Materials	Thickness and bias voltage	kVp for X-ray	Response / decay time	Reference
Cs ₂ AgBiCl ₆	2 V	30	< 3 ms	10.1021/acsaelm.2c00752
Cs ₂ AgBiBr ₆	3 V mm ⁻¹	50	0.77 ms	10.1038/s41566-017-0012-4
(MA) ₃ Bi ₂ I ₉	60 V mm ⁻¹	40	23.3 / 31.4 ms	10.1016/j.matt.2020.04.017
(MA) ₃ Bi ₂ I ₉	286 V mm ⁻¹	40	0.266 / 0.417 ms	10.1002/adom.202000814
(NH ₄) ₃ Bi ₂ I ₉	5 V mm ⁻¹	8	5 ms	10.1038/s41566-019-0466-7
(BDA)PbI ₄	310 V mm ⁻¹	40	7.3 / 22.5 ms	10.1002/anie.202004160
(BA) ₂ PbBr ₄	0.2 V	50	307 / 98 ms	10.1002/eem2.12487
(BA) ₂ PbI ₄	10 V mm ⁻¹	30	4.5 / 4.3 ms	10.1021/acsp Photonics.2c00776
(FA) _{0.55} (MA) _{0.45} PbI ₃	0 V	50	0.086 / 0.5 ms	10.1002/adfm.202109149
MAPbI ₃			0.98 / 1.02 ms	
MAPbBr ₃	0.5 V mm ⁻¹	8	0.255 ms	10.1038/nphoton.2017.43

MAPbBr ₃	0.05 V mm ⁻¹	22	0.73 ms	10.1038/nphoton.2016.41
CsFAGA:Sr			880 / 750 ms	
CsFAGA	0.1 V mm ⁻¹	60	940 / 870 ms	10.1038/s41566-022-01024-9
CsFA			1120 / 1070 ms	
Ours, CsPbBr ₃	0.4 V mm ⁻¹	50	2.89 / 2.50 ms	This work

Table R2. Response time comparison of PSC visible photo-detectors.

Materials	Thickness and bias voltage	Light source	Response / decay time	Reference
FA _{0.55} MA _{0.45} PbI ₃	1 V	870 nm	34 / 164 μs 1.7 / 3.9 μs 23 / 19 μs	10.1039/C7TA04608A
(BA) _n (MA) _{n-1} Pb _n I _{3n+1}	1 V	Xenon lamp	0.25 / 1.75 ms 383 / 177 ms 773 / 385 ms	10.1021/acsnano.8b01999
CsPbBr ₃	0 V	550 nm	230 / 60 ms	10.1002/adom.201600704
	60 V mm ⁻¹		470 / 530 μs	
MAPbI ₃	100 V mm ⁻¹	532 nm	420 / 470 μs	10.1002/adma.202103078
	140 V mm ⁻¹		420 / 460 μs	
MAPbBr ₃	@ 0 V		8 / 613 μs	
MAPbCl ₃	450 μm @ 1 V	355 nm	25 / 940 μs	10.1002/adom.202200449
MAPbBr ₃	@ 1 V		19 / 860 μs	
(BA) ₂ (MA) ₂ Pb ₃ I ₁₀		365 nm	0.5 / 20-60 μs 410 / 310 μs	10.1002/adom.202101145
(PEA) ₂ PbBr ₄	72 V mm ⁻¹	380 nm	0.147 / 0.768 μs	
LDSC-MAPbBr ₃	2.02 mm @ 0 V	White light	93 μs	10.1038/s41467-020-15037
	@ 4 V		62 μs	
HT-MAPbBr ₃	1.42 mm @ 0 V	LED	206 μs	-x
	@ 4 V		205 μs	
MAPbBr ₃	1 mm @ 0.1 V	470 nm	216 μs	10.1038/NPHOTON.2016.
	2.6 mm		1300 μs	41
Ours, CsPbBr ₃	0.4 V mm ⁻¹	365	40 / 710 μs	This work

The dynamic process of the device interface is as follows:

① X-ray photons follow the following physical process. Electron and hole carriers are generated in the perovskite PSC, simultaneously.

Interaction pathway	Products (participating in the cycle)	
	Primary products	Secondary products
Photoelectric effect	Characteristic X-ray	Bremsstrahlung and ionization excitation (leading δ-ray)
	Auger electron	
	Photo-electron	
Compton	Characteristic X-ray	

effect	Scattered photon Compton recoil electron Auger electron	Bremsstrahlung and ionization excitation (leading δ -ray)
Coherent scattering	Scattered photon	
Pair production (requires a high level of energy)		
Photo-nuclear reaction (requires a high level of energy)		

② Driven by the external electric field, carriers separate, drift, and are collected at the electrodes on both sides.

③ The periodic arrangement of the crystal lattice is interrupted at the interface, which creates defects. If deep defects exist at the interface, they trap carriers and allow them to recombine there. The photo-response signal that the electrode can collect becomes smaller.

④ If shallow defects exist on the interface, they capture one type of carriers and release them after a while, delaying the directional drift under the external electric field. The electrode on one side injects carriers to ensure that the collected charge is neutral. That is, the detector develops gain.

3. Please show a comparison chart of perovskite X-ray detectors with your works including active materials, power consumption, device size, LDR, sensitivity, stability, response time, array size, etc to highlight your work.

Reply: Thanks for your suggestion.

We have shown the comparison chart of semiconductor perovskite X-ray detectors in **Table R3** as below. The articles reported in it, is the same as those used in Figure 3e.

Table R3. Comparison chart of perovskite X-ray detector.

Active layer	Intelligent imaging	CMOS-integrated	LDR (dB)	Sensitivity ($\mu\text{C Gy}_{\text{air}}^{-1} \text{cm}^{-2}$)	Size	Response time	Stability	Reference
(NH ₄) ₃ Bi ₂ I ₉	×	×	20	Parallel: 8.4×10^4 Perpendicular: 803	SC: 1~2 cm	Parallel: 13 ms Perpendicular: 5 ms	NA	10.1038/s41566-019-0466-7
Cs ₃ Bi ₂ I ₉	×	×	20	1652	Image: 7.5 cm SC: 0.5-1 mm ²	NA ^a	Keep stable under bias and radiation	10.1038/s41467-020-16034-w
FPEA SC	×	×	26	3402	SC: 1×1 cm ²	0.8 μs , fitting the transient photocurrent	Keep stable under bias and radiation	10.1002/adma.202003790
BAMA film	×	×	84	0.276×10^6	NA	0.5 / 20-60 μs	Keep stable under bias and radiation	10.1126/sciadv.aay0815
2D/3D film	×	×	71	1.95×10^4	Image: 5.1×5.1 cm ²	X-ray: 216 / 174 ms 233 / 182 ms	Keep stable under bias and radiation	10.1002/advs.202102730
MAPbBr ₃ SC	×	×	36	8.4×10^4	SC: 1×1 cm ²	390 nm: 23 μs	Keep stable	10.1038/nmat4927
MAPbBr ₃ SC	×	×	77	2.1×10^6	Image: 1.5 cm SC: 5.8×5.8 mm ²	255 μs	NA	10.1038/nphoton.2017.43
CsPbI ₂ Br film	×	×	24	1.2×10^6	12×12 cm ²	X-ray: 769 ns (1 μm)	Keep stable under radiation	10.1021/acsami.2c03114
CsPbBr ₃ SC	×	×	90	15 to 5111 (depends)	Ingot: 1 inch SC: 5×5 mm ²	γ : 2.6 μs @ 300 V	Keep stable under bias and radiation	10.1002/adfm.202112925
CsPbBr ₃ SC	√	√	106	396	Ingot: 1 inch SC: 5×5 mm ²	X-ray: 2.89 / 2.50 ms 365 nm: 30 / 710 μs γ : 4 μs @ 800 V	Keep stable under bias and radiation	This work

(a) NA means “not applicable”.

4. In Figure S2, If the thickness of C₆₀ is 6 nm, and the roughness is 2.84 nm, then the black spots in Fig. S2c are regarded as pinhole defects, which can be trap sites and degrade the device performances (<https://pubs.acs.org/doi/10.1021/acsenerylett.0c02573>). How can we verify the defect-free device by C₆₀ passivation by those results?

Reply: Thanks for your question.

We didn't make it clear in the previous version. These are not “pinhole defects”. From the results in **Figure R4**, the “spots” in the AFM image have a height of approximately 20 nm and appear brighter in the SEM image, indicating they are taller. They have widths of several tens of nanometers. We speculate that this originates from the reunion of C₆₀ during the evaporation and deposition process.

We have added images showing this CsPbBr₃ sample without C₆₀ in adjacent areas, or rather, the CsPbBr₃ before the deposition of C₆₀, which exhibits extremely high flatness and low roughness. In contrast, the areas with C₆₀ show slight differences in morphology.

Figure R4. Characterization of the sample with or without the C₆₀ layer. (a) Sample photos taken by mobile phones, optical microscopes, and fluorescence microscopes. (b-c) are figures for the areas with C₆₀ only in some places and without C₆₀ in others. (d-g) are figures for the areas with C₆₀ all over the PSC. Figure g shows a certain row of data in Figure f. (h-i) are figures for the areas without C₆₀.

Minor comments:

1. Please add the scale bar in Figures S2a and b. In addition, Please change Fig. S2a to the top-view photo of the CsPbBr₃ single crystal with a C₆₀ passivation layer.

Reply: Thank you for your suggestion.

We have provided scale bars, or length information, for all images containing Figure S2 and R1-4. We have also followed your suggestions and retaken photos of the samples.

2. *In Figure S8, please change the reference form from DOI to reference number.*

Reply: Thank you for your suggestion. We have revised the information for the reference.

REVIEWERS' COMMENTS

Reviewer #3 (Remarks to the Author):

The authors have addressed most of my questions. It might be acceptable now.